# Effect of Acute and Chronic Oral l-Carnitine Supplementation on Exercise Performance Based on the Exercise Intensity: A Systematic Review

**DOI:** 10.3390/nu13124359

**Published:** 2021-12-03

**Authors:** Juan Mielgo-Ayuso, Laura Pietrantonio, Aitor Viribay, Julio Calleja-González, Jerónimo González-Bernal, Diego Fernández-Lázaro

**Affiliations:** 1Department of Health Sciences, Faculty of Health Sciences, University of Burgos, 09001 Burgos, Spain; jjgonzalez@ubu.es; 2Faculty of Sport Science, Universidad Europea de Madrid, 28670 Madrid, Spain; laurampm95@gmail.com; 3Glut4Science, Physiology, Nutrition and Sport, 01004 Vitoria-Gasteiz, Spain; aitor@glut4science.com; 4Department of Physical Education and Sport, Faculty of Education and Sport, University of the Basque Country, 01007 Vitoria, Spain; julio.calleja.gonzalez@gmail.com; 5Department of Cellular Biology, Histology and Pharmacology, Faculty of Health Sciences, Campus of Soria, University of Valladolid, 42003 Soria, Spain; diego.fernandez.lazaro@uva.es; 6Neurobiology Research Group, Faculty of Medicine, University of Valladolid, 47005 Valladolid, Spain

**Keywords:** lipid oxidation, aerobic performance, anaerobic performance, recovery, sports nutrition

## Abstract

l-Carnitine (l-C) and any of its forms (glycine-propionyl l-Carnitine (GPL-C) or l-Carnitine l-tartrate (l-CLT)) has been frequently recommended as a supplement to improve sports performance due to, among others, its role in fat metabolism and in maintaining the mitochondrial acetyl-CoA/CoA ratio. The main aim of the present systematic review was to determine the effects of oral l-C supplementation on moderate- (50–79% $\dot{V}$O_2 max_) and high-intensity (≥80% $\dot{V}$O_2 max_) exercise performance and to show the effective doses and ideal timing of its intake. A structured search was performed according to the PRISMA^®^ statement and the PICOS guidelines in the Web of Science (WOS) and Scopus databases, including selected data obtained up to 24 October 2021. The search included studies where l-C or glycine-propionyl l-Carnitine (GPL-C) supplementation was compared with a placebo in an identical situation and tested its effects on high and/or low–moderate performance. The trials that used the supplementation of l-C together with additional supplements were eliminated. There were no applied filters on physical fitness level, race, or age of the participants. The methodological quality of studies was evaluated by the McMaster Critical Review Form. Of the 220 articles obtained, 11 were finally included in this systematic review. Six studies used l-C, while three studies used l-CLT, and two others combined the molecule propionyl l-Carnitine (PL-C) with GPL-C. Five studies analyzed chronic supplementation (4–24 weeks) and six studies used an acute administration (<7 days). The administration doses in this chronic supplementation varied from 1 to 3 g/day; in acute supplementation, oral l-C supplementation doses ranged from 3 to 4 g. On the one hand, the effects of oral l-C supplementation on high-intensity exercise performance variables were analyzed in nine studies. Four of them measured the effects of chronic supplementation (lower rating of perceived exertion (RPE) after 30 min at 80% $\dot{V}$O_2 max_ on cycle ergometer and higher work capacity in “all-out” tests, peak power in a Wingate test, and the number of repetitions and volume lifted in leg press exercises), and five studies analyzed the effects of acute supplementation (lower RPE after graded exercise test on the treadmill until exhaustion and higher peak and average power in the Wingate cycle ergometer test). On the other hand, the effects of l-C supplementation on moderate exercise performance variables were observed in six studies. Out of those, three measured the effect of an acute supplementation, and three described the effect of a chronic supplementation, but no significant improvements on performance were found. In summary, l-C supplementation with 3 to 4 g ingested between 60 and 90 min before testing or 2 to 2.72 g/day for 9 to 24 weeks improved high-intensity exercise performance. However, chronic or acute l-C or GPL-C supplementation did not present improvements on moderate exercise performance.

## 1. Introduction

Athletes often turn to nutritional supplements in order to maintain health and maximize athletic performance [1]. Among them, as a supplement, l-Carnitine (l-C) became popular following rumors that it helped the Italian national soccer team to win the world championship in 1982, and it is often portrayed as a “fat burner”, supposedly by increasing the aerobic contribution to exercise by increasing fat oxidation and muscle mass and reducing fat mass [2]. Carnitine is a dipeptide which is an essential factor for the membrane transport of acyl-coenzyme A (CoA) and is involved in the oxidation of fatty acids in the liver and kidney [3]. For that reason, the main hypothesis of the cited studies was that the potential increase in carnitine concentration in skeletal muscle would lead to an increase in fatty-acid transport and oxidation, improving oxygen consumption ($\dot{V}$O_2_) and endurance performance [4]. Likewise, in high-intensity exercise, carnitine would reduce the blood lactate accumulation since it reacts with excess acetyl-CoA and forms acetyL-Carnitine and CoA [5]. Moreover, carnitine may enhance blood flow and oxygen supply to the muscle tissue via improved endothelial function, thereby reducing hypoxia-induced cellular and biochemical disruptions and, therefore, improving muscle recovery [6]. Likewise, the antioxidant effect of carnitine by increased overall antioxidant enzyme status, could be effective on muscle recovery [7]. For all these reasons, l-Carnitine (l-C) supplementation has shown potential multiple effects on different physiological and metabolic pathways that could improve athletic performance in both moderate- and high-intensity exercises (Figure 1) [8]. These mechanisms have been hypothesized to impact sports performance through increasing and maintaining a high muscle carnitine content [9]. Nonetheless, it is difficult to increase muscle content due to the elevated transmembrane gradient for l-C through the skeletal muscle, which facilitates carnitine output from muscle to the plasma [10]. Therefore, l-C oral supplementation or another supplement with l-C (glycine-propionyl l-Carnitine (GPL-C) or l-Carnitine l-tartrate (l-CLT)) could be an effective strategy to achieve an increase in carnitine content in the muscle [11].

Regarding high-intensity exercise performance (≥80% $\dot{V}$O_2 max_) [12], the role of carnitine in maintaining the acetyl-CoA/CoA ratio is especially important in maximum and supramaximal exercises, such as sprints competitions or mid-distance races between 50 and 400 m in swimming [13]. In fact, previous studies have reported that long-term oral l-C supplementation (2 g/day for 4 weeks) produced certain improvements on sports performance determinants such as $\dot{V}$O_2 max_ (mL/min/kg) and maximal work output (KJ) in both elite and amateur athletes [2]. In contrast, other studies did not find improvements in other parameters such as the speed achieved during high-intensity interval exercise (HIIT) in swimming [14] or in the consumption of oxygen, maximum power, or work done during HIIT exercise [15]. On the other hand, some investigations reported that acute oral l-C supplementation improved the speed achieved and rating of perceived exertion in incremental tests until exhaustion after the intake of 3 or 4 g l-C combined with CHO 1 h before testing [16]. 

On the other hand, the first oral l-C supplementation studies failed to obtain improvements on moderate-intensity exercise performance (50–79% $\dot{V}$O_2 max_) [12], measured as changes in heart rate responses during different cycle tests at 50% $\dot{V}$O_2 max_ after supplementation with 1 g/day for 14 or 28 days [17] or 5 g/day for 5 days [18]. Likewise, Soop et al. did not show differences in substrate oxidation or blood lactate concentration during prolonged moderate intensity cycling exercise [18]. These results may have been because the duration and/or dose was insufficient to increase the muscle carnitine pool and fatty-acid oxidation during prolonged exercise [19,20]. However, Stephens et al. demonstrated with muscle biopsies that an insulin stimulus, by using insulin intravenous infusions [21,22] or CHO intake [9,10], increased carnitine of muscle content after oral L-C supplementation. In this line, Wall et al. presented positive results regarding muscle glycogen saving during moderate-intensity exercise, which produced an increase in working capacity and a decrease in the perceived exertion scale in subsequent tests of greater intensity after a chronic supplementation of 4 g/day of oral l-C supplementation together with 80 g/day CHO (24 weeks) [23]. On the other hand, in studies with acute oral l-C supplementation protocols (2–3 g of oral l-C 2–3 h before exercise) together with CHO intake, no changes were reported in the use of substrates; hence, the increase in muscle carnitine pool was seemingly not enough to produce improvements in this type of exercise performance [20,24]. These results could indicate that, in addition to the content of the muscle carnitine pool, there could be other mechanisms via which l-C could be effective on sports performance.

Therefore, although oral l-C or glycine-propionyl l-Carnitine (GPL-C) or l-Carnitine l-tartrate (l-CLT) supplementation could be effective in both high- and moderate-intensity sports performance, the current scientific evidence is controversial. In this sense, to the best of author’s knowledge, there is no clear information regarding the different effects involved in high- and moderate-intensity performance, which are needed to understand whether oral l-C supplementation can be effective. For that reason, the main aim of the present systematic review was to determine the effects of oral l-C, glycine-propionyl l-Carnitine (GPL-C), or l-Carnitine l-tartrate (l-CLT) supplementation on high- and moderate-intensity exercise performance, as well as to identify the effective doses and ideal timing of their intake.

## 2. Materials and Methods

### 2.1. Literature Search Strategies

The current systematic review was conducted on the basis of specific methodological guidelines of Preferred Reporting Items for Systematic Review and Meta-Analysis (PRISMA) [25] and the PICOS question model for the definition of inclusion criteria: P (population): “both physically active and untrained people”, I (intervention): “l-Carnitine supplementation”, C (comparison): “same conditions with placebo or control group”, O (outcome): “sports performance variables”, S (study design): “double-blind and randomized parallel or crossover design” [26]. The systematic search of the current scientific literature was undertaken for scientific articles that analyzed the effects of l-Carnitine supplementation on high-intensity exercise performance (≥80% $\dot{V}$O_2 max_) and moderate-intensity (50–79% $\dot{V}$O_2 max_) exercise performance [12]. These cutoff points of the exercise intensity were based on the study by Trinity et al., where they grouped the exercise intensities as high intensity such as 80–90–100% $\dot{V}$O_2 max_ and moderate intensity such as 50–60–70% $\dot{V}$O_2 max_. The search was carried out in the Scopus and Web of Science (WOS) databases, considering that the latter includes other databases such as CCC, DIIDW, KJD, MEDLINE, RSCI, and SCIELO. Results obtained up to 24 October 2021 were included, but no filter was applied to narrow the search by year. The following search Boolean equation was used to find the relevant articles: (“carnitine”[All Fields] OR “carnitine”[MeSH Terms] OR “carnitine”[All Fields] OR “carnitine s”[All Fields] OR “carnitines”[All Fields] OR (“carnitine”[MeSH Terms] OR “carnitine”[All Fields] OR “l carnitine”[All Fields])) AND (“supplemental”[All Fields] OR “supplementing”[All Fields] OR “supplementation”[All Fields] OR “supplementation s”[All Fields] OR “supplementations”[All Fields] OR “supplementation”[All Fields]) AND (“sport s”[All Fields] OR “sports”[MeSH Terms] OR “sports”[All Fields] OR “sport”[All Fields] OR “sporting”[All Fields] OR (“athlete s”[All Fields] OR “athletes”[MeSH Terms] OR “athletes”[All Fields] OR “athlete”[All Fields] OR “athletically”[All Fields] OR “athletes”[All Fields] OR “sports”[MeSH Terms] OR “sports”[All Fields] OR “athletic”[All Fields] OR “athletics”[All Fields]) OR (“exercise”[MeSH Terms] OR “exercise”[All Fields] OR “exercises”[All Fields] OR “exercise therapy”[MeSH Terms] OR (“exercise”[All Fields] AND “therapy”[All Fields]) OR “exercise therapy”[All Fields] OR “exercise s”[All Fields] OR “exercised”[All Fields] OR “exerciser”[All Fields] OR “exercisers”[All Fields] OR “exercising”[All Fields])) AND (“perform”[All Fields] OR “performable”[All Fields] OR “performance”[All Fields] OR “performance s”[All Fields] OR “performances”[All Fields] OR “performative”[All Fields] OR “performatively”[All Fields] OR “performatives”[All Fields] OR “performativity”[All Fields] OR “performativity”[All Fields] OR “performed”[All Fields] OR “performer”[All Fields] OR “performer s”[All Fields] OR “performers”[All Fields] OR “performing”[All Fields] OR “performs”[All Fields]). No filters were applied to the athlete’s physical fitness level, race, or age to increase the power of the analysis. Through this equation, related scientific articles in this area were obtained applying the snowball strategy which returned relevant articles in the field [27].

All titles and abstracts resulting from the search were referenced to identify duplicates and potential missing studies. Titles and abstracts were selected for subsequent full-text review. After the identification of potentially relevant articles, the full text was read, evaluating their eligibility after evaluation with the designated inclusion criteria. The search for published articles was carried out independently by two different authors (J.M.-A. and L.P.). In case of disagreements related to the main outcomes, decisions were finalized through third author (J.C.-G.).

### 2.2. Inclusion and Exclusion Criteria

The articles obtained in the search by the following inclusion criteria were followed to select the studies used in the final review: (i) original articles with double-blind and crossover or parallel controlled design; (ii) the impact of supplementation with l-C or GPL-C or l-CLT in known amounts administered before and during exercise in healthy humans was investigated; (iii) with identical experimental conditions where a placebo was administered or there was a control group; (iv) at least one of the variables measured was on high- (≥80% $\dot{V}$O_2 max_) and moderate-intensity (50–79% $\dot{V}$O_2 max_) exercise performance [12]; (v) with clear information on the administration of supplementation (doses and timing intake); (vi) published in any language.

On the other hand, the exclusion criteria applied were the following: (i) studies carried out on animals or sick participants; (ii) studies in which l-C was part of a compound with multiple supplements; (iii) studies whose variables were not related to human performance; (iv) studies in which the results or the method used was not clear; (v) uncontrolled trials; (vi) studies carried out using participants with cardiovascular, metabolic, or musculoskeletal disorders.

Once the inclusion/exclusion criteria were applied to each study, data on study source (including authors and year of publication), study design, supplement administration (dose and timing), sample size, characteristics of the participants (level, race, and gender), and final outcomes of the interventions were extracted independently by two authors (J.M.-A. and L.P.) using a spreadsheet (Microsoft Inc, Seattle, WA, USA). Subsequently, disagreements were resolved through discussion until a consensus was reached or through third-party adjudication (J.C.-G.).

### 2.3. Study Selection

Two authors identified papers through a database search. The titles and abstracts of publications identified by the search strategy were screened for a subsequent full-text review and were cross-referenced to identify duplicates. All trials assessed for eligibility and classified as relevant were retrieved, and the full text was peer-reviewed (L.P and J.M-A). Moreover, the reference sections of all relevant articles were also examined by applying the snowball strategy [27]. On the basis of this information within the full reports, inclusion and exclusion criteria were used to detect the potential studies eligible for inclusion in this systematic review. Disagreements were resolved through discussions among the different authors.

### 2.4. Outcome Measures

The scientific literature was recollected related to the effects of oral l-C, GPL-C, or l-CLT supplementation on both high- (≥80% $\dot{V}$O_2 max_) and moderate-intensity (50–79% $\dot{V}$O_2 max_) exercise performance, as well as to identify the potential effective doses and ideal timing of their intake. These results could be influenced by type of sport, dose of each supplementation and duration of the intervention, and participant characteristics, such as age, gender, ethnicity, body composition, training level, differences in training, nutrition and health status, or ethnicity.

### 2.5. Quality Assessment of the Experiments

The methodological quality and risk of bias were assessed by two authors independently (L.P and J.M-A), and disagreements were resolved by third party evaluation (J.C-G), in accordance with the Cochrane Collaboration Guidelines [28]. These selected studies were assessed using the McMaster Critical Review Form for Quantitative Studies [29]. This evaluation was used to determine the methodological limitations existing in each of the studies, thus allowing the quality of the results to be comparable among the different study designs. The form consists of 16 items that evaluate criteria such as clarity in the purpose of the study (item 1), the relevance of the literature used (item 2), the aptitude of the study design (item 3), sample described in detail (items 4 and 5), use of informed consent (item 6), reliability and validity of the outcome measures (items 7 and 8), description of methods (item 9), statistical significance of the results (item 10), adequate analysis methods (item 11), the importance of informed practice (item 12), dropout report (item 13), adequacy of the conclusions (item 14), practical implications (item 15), and limitations recognized by the authors (item 16). Criteria allow for an answer “yes = 1 point” or “no = 0 points”, except for items 6 and 13 that present the option “if it is not applicable, suppose 3”, entered to eliminate the negative effect of assuming the value 0 on a binary scale when that specific item did not apply to that study. The scoring scale is divided into five quality categories: poor methodological quality (≤8 points), acceptable methodological quality (9 to 10 points), good methodological quality (11 to 12 points), very good methodological quality (from 13 to 14 points), and excellent methodological quality (≥15 points).

The scores ranging from 9–14 points were obtained (Table 1), representing a minimum methodological quality of 60% and a maximum of 93.33%. Out of the 11 studies, three achieved acceptable methodological quality [14,15,20], two achieved good methodological quality [30,31], and the other six studies achieved very good quality [16,23,24,32,33,34]. No study was excluded for not meeting the minimum quality threshold. The table details the results of the evaluated criteria, where the main deficiencies were found in the methodological quality. These were mainly associated with items 5 and 7 of the questionnaires corresponding to the detailed justification about the sample size and the reliability of study results, respectively.

## 3. Results

### 3.1. Main Search

After applying the previous search equation, 216 articles were identified through database searches, and four additional articles identified from other sources. After the elimination of duplicate articles (*n* = 51), 171 articles were examined by title and abstract. Among these studies, 151 were removed with the following reasons: 48 were non-intervention studies, 37 used animals or sick people as test participants, 15 studied variables that did not occupy the main aim of this systematic review, and six administered l-C mixed with other supplements. A full-text evaluation was performed on the remaining 65 articles applying the inclusion criteria. Out of the 65 articles, 30 were eliminated for not using the double-blind design, 23 were eliminated for not having at least one outcome related to sports performance, and one was eliminated for not having clear information about the administration of supplementation. Therefore, 11 articles were finally included in the current systematic review [14,15,16,20,23,24,30,31,32,33,34] (Figure 2).

### 3.2. Carnitine Supplementation

Participants and intervention characteristics of the studies included in this systematic review are described in Table 2. The total sample consisted of 203 participants (men = 180; women = 23), and all studies were carried out with adult population aged between 18 and 46 years. In all cases, both physically active and untrained participants were selected to carry out the studies. Out of those, only two of them selected professional participants [16,31], seven studies were carried out with recreational or amateur athletes [14,20,23,24,30,32,33], and two of them involved participants without any kind of previous training [15,34].

Regarding the type of l-Carnitine supplemented, six studies used l-C [14,16,20,24,31,33] while three articles used l-CLT [15,23,30] and two others combined the molecule propionyl l-Carnitine (PL-C) with GPL-C [32,34].

Regarding supplementation time, five studies analyzed the chronic supplementation of l-C with a duration (depending on the study) ranging between 4 and 24 weeks [15,23,30,33,34]. The other six studies used an acute administration for 7 days, before and/or during the test [14,16,20,24,31,32]. Specifically, the administration times varied among the studies in a range between 1 and 3 h before the test, while only one study administered it during the test [20], in which participants of a marathon race were supplemented 2 h before testing and 20 km after starting.

Lastly, the administration doses in this chronic supplementation varied from 1 to 3 g/day depending on the study; however, in the case of one of the studies, two different doses were analyzed (1 or 3 g/day) [34]. However, in acute supplementation, l-C supplementation doses ranged from 3 to 4 g.

### 3.3. High-Intensity and Moderate-Intensity Exercise Performance Outcomes Analyzed

The studies included in this systematic review measured a wide range of variables. Consequently, the studies were divided by the nature of their outcomes into two different tables. The first one (Table 3) includes the studies related to high-intensity performance variables [15,16,23,24,30,31,32,33,34], while the second (Table 4) includes the studies with moderate-intensity performance variables [20,23,24,30,31,34].

#### 3.3.1. Effect of l-Carnitine Supplementation on High-Intensity Exercise Performance (≥80% $\dot{V}$O_2 max_)

The effects of l-C supplementation on high-intensity exercise performance variables were analyzed in nine studies [15,16,23,24,31,32,33,34]. Out of those, four measured the effects of chronic supplementation [15,23,33,34] and five measured the effects of acute supplementation [16,24,31,32,35], all of them compiled in Table 3.

Firstly, for studies that used chronic l-C supplementation, Shannon et al. [15] measured variables in untrained participants after 24 weeks of supplementation and HIIT training, such as $\dot{V}$O_2 max_, maximum power, and total work in a high-intensity interval test on a cycle ergometer, without finding any significant differences in any parameters. On the other hand, Wall et al. [23] used 2 g/day of l-CLT (1.36 g of l-C) supplementation and carried out the same 24-week l-C supplementation period as Shannon et al. [15]; however, in this case, the participants analyzed were recreational athletes. A 30 min test was carried out at 80% $\dot{V}$O_2 max_, followed by another 30 min “all out” test using a cycle ergometer, and significant improvements were found in both cases on perceived exertion by Borg scale and work output. Koozehchian et al. [33] carried out in their study different high-intensity tests in endurance-trained athletes. Significant improvements were found in peak power and absolute peak power (30 s Wingate test) after 9 weeks of 2 g/day l-C supplementation. Additionally, they showed improvements on the number of repetitions and volume lifted in the third set of leg press [33]. Despite performing the same cycle ergometer test, Smith et al. [34] did not observe significant differences in high-intensity power parameters (peak and average power), in percentage of fatigue, and in total work (kJ), after 8 weeks of supplementation. In their case, the test was performed on untrained participants, and they used different types of supplementation: 1 g/day of PL-C and 3 g/day, compared to the 2 g/day of l-C that Koozehchian et al. [33] used in their study. This type of supplementation could be one of the possible causes of the difference found in their results.

A further five studies measured the effect of acute supplementation with l-C on high-intensity performance variables [16,24,31,32,35]. In the first study, Trappe et al. (1994) [14] performed five repeated swims of 91.4 m (100 yd) at a supramaximal intensity with 2 min rest between each. These authors did not observe statistical differences between groups in swimming velocity after 4 g/day of l-C in a citrus drink twice daily for 7 days. Jacobs et al. [32] supplemented 4.5 g of GPL-C (3 g PL-C) with 8 oz of water 90 min before the Wingate test without previous CHO load. Significant improvements were found in the peak and mean power of the last series carried out by the participants. In the study carried out by Orer and Guzel [16], two different l-C supplementation doses (3 and 4 g/day) were prescribed with a glass of fruit juice in professional soccer players, 1 h before carrying out an incremental test on a treadmill until exhaustion. Differences between supplements were not significant, and, after their ingestion, the results showed how the participants reached higher speeds and better data in perception of effort, suggesting that acute supplementation with l-C could delay fatigue. Arazi and Mehrtash showed significant differences in maximum power in an anaerobic sprint test (RAST) after 3 g of l-C 90 min before the test [31]. Moreover, these authors showed significant differences in $\dot{V}$O_2 max_ after a 20 m shuttle run in 18 male artistic gymnasts after l-C supplementation. Lastly, Burrus et al. performed a cycle ergometer test at 85 % $\dot{V}$O _2max_ until exhaustion was reached [24]. During this process, the moderately trained participants ingested 3 g of l-C supplementation with a CHO load (3 h before competition), without statistical differences in power achieved and time to exhaustion.

#### 3.3.2. Effect of l-Carnitine Supplementation on Moderate-Intensity (50–79% $\dot{V}$O_2 max_) Exercise Performance

The effects of l-C supplementation on moderate-intensity exercise performance variables were analyzed in four studies [20,24,30,34]. Out of those, two measured the effect of a chronic supplementation [30,34], and three of them described the effect of an acute supplementation [20,24]. All studies are described in Table 4.

The studies with chronic supplementation doses of l-C varied between 1 and 3 gr of dose, but it should be noted that only Wall et al. [23] prescribed the joint intake of CHO with supplementation. In their case, the study presented a longer duration (24 weeks) than the study carried out by Smith et al. [34] and Broad et al. [30], which were 8 and 4 weeks, respectively. None of them found significant improvements in any of the parameters studied to measure moderate-intensity exercise performance. The tests carried out to measure these variables were a 30 min test at 50% $\dot{V}$O_2 max_ [23] and a 90 min test at 65% $\dot{V}$O_2 max_ [30], both tested using a cycle ergometer. However, in the study carried out by Smith in 2008, the authors used a gradual exercise test after 8 weeks of aerobic training in order to measure these parameters together with the total time of exercise, without significant improvements in any case. 

In acute supplementation studies [20,24,31], the intake of l-C along with CHO was prescribed. For example, Colombani (1996) [20] measured the time necessary to finish a marathon race in runners. Participants consumed l-C 2 h before and 20 km after starting. The main results did not find significant improvements in total marathon time between groups. Burrus et al. [24] carried out in their study a similar test to Wall et al. [23], in which, after taking 3 g of LC 3 h prior, the participants were asked to perform 40 min on a cycle ergometer at 65% of $\dot{V}$O_2 max_, but the power reached measured from the beginning and every 10 min did not improve significantly.

## 4. Discussion

The main aim of this systematic review was to determine the effects of l-C supplementation on exercise performance, analyzing the current evidence and evaluating its impact on high- and moderate-intensity exercise performance. In addition, it summarized the effective doses and ideal timing of their intake according to analyzed studies. The main results indicated that specific oral l-C supplementation protocols could result in significant improvements in high-intensity exercise performance. However, no significant changes in low–moderate-intensity performance were found. The studies that achieved improvements in high-intensity (≥80% $\dot{V}$O_2 max_) exercise performance used acute doses of 3 to 4 g of l-C or GPL-C ingested between 60 and 90 min before exercise and chronic doses of 2 to 2.72 g/day of l-C for longer periods up to 9 to 24 weeks.

The amino acid carnitine is well known for its role in the transport of long-chain fatty acids into the mitochondrial matrix, where they are oxidized [8]. Furthermore, by generating acylcarnitines, l-C defends the cell from acyl-CoA accretion [2]. Carnitine is mostly obtained from animal-based foods, with endogenous production in the liver and kidney providing a minor contribution. Due to the muscle’s inability to generate carnitine, it is dependent on uptake from the bloodstream [36]. Mitochondrial fatty-acid oxidation is a significant source of energy for muscle metabolism, especially during exercise [37]. However, the availability of free l-C in the mitochondria appears to be limited during this process, especially during high-intensity exercise [8]. Therefore, fatty-acid oxidation decreases significantly as exercise intensity rises from moderate to high. Given the importance of fatty acids in muscle bioenergetics and the limiting influence of free carnitine on fatty-acid oxidation during endurance exercise, l-C supplementation has been proposed as a means of improving exercise performance [2]. The function of l-C supplementation on muscular performance has not yet been conclusively established. Different experimental outcomes were obtained due to differences in exercise (high and low–moderate) intensity, amount of l-C provided, method, and timing of administration relative to the exercise (acute and chronic). In this review, the role of l-C in muscle energetics was discussed, as well as the primary reasons for contradicting findings on the supplementation of l-C.

### 4.1. Effects of l-C Supplementation on High-Intensity Exercise Performance (≥80% $\dot{V}$O_2 max_)

When exercise intensity is greater than 80% $\dot{V}$O_2 max_, the use of glycogen begins to predominate over fatty acids as an energy source. It represents a key fuel during long-duration (>1 h) and high-intensity exercise [8]. In this sense, during high-intensity exercise, muscle carnitine loading by l-C supplementation could result in a better matching of glycolysis, pyruvate dehydrogenase complex (PDC), and mitochondrial flux, thereby reducing muscle anaerobic energy generation [23]. In this sense, carnitine’s principal physiological roles during high-intensity exercise goes from acetyl group buffering (i.e., generating acetylcarnitine) to maintaining a pool of free co-enzyme A, which is required for mitochondrial flow to proceed (including the pyruvate dehydrogenase complex reaction) [38]. However, there is still a rise in the acetyl-coenzyme A (acetyl-CoA)/CoASH ratio during this type of activity, presumably due to the considerable depletion of the free carnitine pool (to <6 mmol/kg dry muscle) induced by acetylcarnitine synthesis [39].

As a result, an increase in skeletal muscle total carnitine content may provide more effective buffering of acetyl-CoA production during high-intensity exercise, counteracting the increase in the acetyl-CoA/CoASH ratio and enhancing PDC flow and mitochondrial ATP production [23]. This would diminish the contribution of glycolysis and PCr hydrolysis to ATP synthesis, especially during the rest-to-exercise transition, when inertia in mitochondrial ATP production is known to exist at the level of PDC activation and flux [40]. Furthermore, increasing PDC flow during high-intensity exercise is likely to lower muscle lactate generation, potentially improving exercise performance by reducing muscular acidosis [41]. These metabolic actions could have significant implications for anaerobic ATP during high-intensity exercise. In this way, improved muscle strength (number of repetitions and third set lifting volume) [33] and anaerobic performance by 30 s Wingate test (mean and peak power) [32,33], as well as decreased perceived exertion effort and an increase in work production [16,23], has been reported. Moreover, decreasing post-exercise blood lactate levels after graded exercise test on the treadmill were revealed [16]. These data suggest that both chronic and acute l-C supplementation could result in improvements in high-intensity performance [16,23,32,33].

#### 4.1.1. Chronic l-C Supplementation Protocol for High-Intensity Exercise Performance (≥80% $\dot{V}$O_2 max_)

According to the results of included studies, 2 to 2.72 g/day of l-C for 9 to 24 weeks could be beneficial in improving athletic performance during high-intensity exercise. Wall et al. [23] revealed that supplementation with 4 g of l-CLT (2.72 g of l-C) with 160 g of CHO distributed two times per day for 24 weeks led to a 35% improvement in work capacity in an “all out” test. This test was performed for 30 min at 50% $\dot{V}$O_2 max_, followed immediately by 30 min at 80% $\dot{V}$O_2 max_ [23]. Authors indicated that this advantage in the “all out” test was because the group supplemented with carnitine had 71% more muscle glycogen compared to the control, particularly due to savings in the 30 min at 50% $\dot{V}$O_2 max_ test [23]. On the other hand, Koozehchian et al. [33] described improvements in peak power and absolute mean power derived from a lower accumulation of blood lactate concentration during a 30 s Wingate test after 2 g/day of l-C supplementation for 9 weeks. In addition, these authors observed improvements in the number of repetitions and lifting volume in the third set of leg press exercises with the same previous protocol [33]. These improvements were attributed to the fact that the increase in training volume was greater in the supplemented group, since the training program intensity was moderate. Therefore, chronic l-C supplementation could result in a greater fatty-acid oxidation rate, thereby preserving muscle glycogen stores [23]. To support this idea, these studies found that increasing muscle carnitine content saves muscle glycogen during low-intensity exercise (consistent with an increase in muscle lipid utilization), but that increasing muscle carnitine content reduces muscle anaerobic ATP production during high-intensity exercise by better matching glycolytic, pyruvate dehydrogenase complex, and mitochondrial flux [23]. Moreover, this effect could be due to decreased post-exercise lactate levels and attenuated exercise-induced oxidative stress markers [33,42].

On the contrary, other studies with chronic supplementation protocols using lower or similar l-C doses (1 to 3 g/day) for periods between 4 and 24 weeks did not obtain improvements in this type of performance [15,30,34]. Smith et al. [34] did not show significant differences in power, total work, or percentage of fatigue after a test of anaerobic power (30 s on cycle ergometer) when the athletes were supplemented with both 1 g/day of PL and 3 g/day of PL for 8 weeks. These authors attributed these results to inadequate testing methods or to the high degree of variability in participant response. In the same line, Broad et al. [30] did not present improvements in 20 km time trial duration after 3 g/day of l-CLT (2 g/day l-C) for 4 weeks. In this case, the high content of prior muscle carnitine could have led to the participants not oxidizing more fatty acids during the test. In addition, the relatively short time of supplementation did not allow for improvements in these deposits. For this reason, the investigators considered that future studies could analyze carnitine administration along with CHO-rich feedings to induce hyperinsulinemia in order to increase skeletal muscle carnitine content [34].

Lastly, HIIT emerged several decades ago in response to the need for training techniques for athletic events of high intensity and often also of long duration [43]. It is well known that HIIT exercise increases glycolytic and mitochondrial ATP and can induce morphological changes such as a conversion of fiber type morphology or increased fiber type area in fast-twitch fibers [44]. In this sense, after 24 weeks of 2 g/day l-CLT (1.36 l-C) with 80 g/day of CHO supplementation during a HIIT training program, Shannon et al. [15] did not find significant improvements derived from less blood lactate accumulation, such as better $\dot{V}$O_2 max_, power, and work performed. However, it is likely that the duration of exercises in this study (two sets of 3 min) was not enough to allow a plateau of acetyl accumulation, which is normally observed within 10 min at 75–90% VO_2_; thus, the ability of l-C to buffer acetyl groups would be less critical when it comes to finding improvements [15]. The authors explained that increased reliance on nonmitochondrial ATP resynthesis during a second bout of intense exercise is accompanied by increased carnitine acetylation. However, muscle carnitine during 24 weeks of HIIT did not alter this effect, nor did it enhance muscle metabolic adaptation or performance gain compared with HIIT alone [15]. Other studies also found no significant differences in performance parameters during high-intensity tests after chronic supplementation of 4 g of l-C with CHO for 14 and 7 days [14], confirming that the supplementation period was insufficient, the exercise intensity was perhaps too great, and the glycolytic flux exceeded the capacity of the carnitine pool to effectively maintain the acetyl CoA/CoA ratio.

#### 4.1.2. Acute l-C Supplementation Protocol for High-Intensity Exercise Performance (≥80% $\dot{V}$O_2 max_)

Doses from 3 to 4 g of l-C or GPL-C supplementation ingested between 60 and 90 min before exercise have been shown to improve different performance parameters. Lactate threshold values achieved at higher speeds and lower levels of perceived exertion scales during incremental tests until exhaustion were presented in [16]. Equally, an increase in peak and average power of the last 10 s sprints in Wingate tests on a cycle ergometer was identified [32]. These effects could be explained by more effective use of the aerobic energy system with the buffering capacity of acylcarnitines, resulting in a delay in the arrival of the lactate threshold and, therefore, consequent fatigue [8]. To provoke the insulin stimulus that increased the muscle carnitine content, Orer and Guzel prescribed 3 or 4 g/day of l-C supplementation in combination with CHO consumed in orange juice [16]. On the other hand, the type of supplementation used by Jacobs et al. [32] was a combination of the amino acid glycine with the molecule propionyl l-Carnitine in order to increase levels of nitric oxide, although the physiological mechanisms remain to be elucidated to the best of the author’s knowledge. It was in 2007 when the ability of glycine to increase the plasma concentration of nitrites and nitrates was reported. This phenomenon could have the potential effect of improving blood flow to skeletal muscle during exercise [45]. Therefore, the significant improvements in Jacobs et al.’s study were attributed to the vasodilator effects of nitric oxide that caused an improvement in the exchange of nutrients and metabolic products in muscle tissue [32]; hence, this kind of GPL-C supplementation could be an alternative to the use of CHO to increase muscle carnitine content.

However, before discovering the necessity to carry out an insulin stimulus in order to increase muscle carnitine reserves, some studies [9,21,22] found lower accumulation of blood lactate concentration and an increase in $\dot{V}$O_2_ and work capacity after oral l-C supplementation without insulin stimulus, particularly in high-intensity exercises [46,47]. In addition, another study found a decrease in blood lactate accumulation and improvements in power after intravenous supplementation [42]. An initial limitation may be the lack of methodology given that they did not present muscle biopsies, highlighting whether it was possible to increase the content of muscle carnitine. For that reason, the concluded improvement in parameters with l-C supplementation is unfounded [35].

On the contrary, other acute supplementation protocols with similar or lower doses (between 3 and 4 g) ingested earlier (between 2 and 3 h before testing) or during the same test did not obtain improvements in high-intensity exercise performance parameters [20,24]. There was a lack of improvement in power or time to exhaustion during a test at 85% $\dot{V}$O_2_ on a cycle ergometer or in the speed achieved at 4 mmol/lactate in tests of submaximal performance after a marathon race [24]. These results were attributed to the fact that the intake of the supplementation period with CHO was not sufficient to increase the carnitine reserves in muscle and modify the substrate used du [20,24]. Furthermore, in the case of Burrus et al. [24], despite meeting the inclusion criteria (45 mL/kg/min of $\dot{V}$O_2 peak_), participants had great variability in the number of days and the duration of the training sessions. Therefore, it may be that the duration to exhaustion was not affected by the differences in participants’ aerobic practice [24]. For this reason, it is necessary to carry out more research to clarify these contradictions and identify the appropriate way to supplement with l-C acutely.

The variability of the results after both acute and chronic supplementation may be derived from differences in methodological approaches, individual differences among participants, type of stimulus, and lack of muscle biopsy data on muscle carnitine content [30]. Even so, from the studies included in this review, it can be concluded that supplementation with 3 to 4 g of l-C or GPL-C ingested 60–90 min before exercise, as well as doses 2 to 2.72 g with CHO over periods of 9 to 24 weeks, could be beneficial to improve performance in incremental tests, 10 s sprints, Wingate tests, and all out tests. These effects could be better if supplementation with l-C includes CHO.

### 4.2. Effects of l-C Supplementation on Moderate-Intensity Exercise Performance (50–79% $\dot{V}$O_2 max_)

Great cardiovascular capacity and metabolic adaptation are required in moderate-intensity exercise performance to supply the O_2_ needs during exercise [48,49]. Although a genetic predisposition is important to perform this type of discipline, training and nutrition protocols are essential to improve performance in moderate-intensity exercise events [50]. From a metabolic point of view, during exercise at 50% $\dot{V}$O_2 max_ intensities or below, fatty-acid oxidation is the most predominant source of energy, whereas, during exercise of moderate intensity (between 50 and 70% $\dot{V}$O_2 max_), the energy source is provided by both fatty acids and glucose, with a gradual increase in the intensity [51].

In this sense, the main role of carnitine during the exercise of low intensity, when the PDC activation and the flow are minimum, very likely involves the translocation of mitochondrial fatty acids [23]. Although it has been suggested that free carnitine only limits fat oxidation at exercise intensities greater than 70% $\dot{V}$O_2 max_, it has also been observed that an acute increase of 15% in the muscle carnitine content reduced glycolytic flow mediated by insulin and PDC activation compared to a control group [22]. In addition, this effect was also followed by an increase in muscle glycogen and long-chain CoA, which indicates an increase in muscle fatty-acid oxidation and CHO storage mediated by carnitine [22]. Therefore, free carnitine availability can be a limiting factor in mitochondrial fatty-acid translocation both at rest and during low-intensity exercise, and an increase in skeletal muscle total carnitine content would increase the fatty-acid oxidation while reducing PDC activation and glycogen utilization during low–moderate-intensity exercise [23]. In this sense, the increase in muscle carnitine increased fat oxidation during low–moderate-intensity exercise could save glycogen reserves and, therefore, decrease the fatigue at intensities below blood lactate threshold [13]. Lastly, another study looked at the effect of supplementation on aerobic capacity, fat oxidation, and $\dot{V}$O_2 max_, i.e., physical performance [42]. Regarding the dose, differences were observed among acute and chronic supplementation protocols, although this review did not find significant improvements in moderate-intensity exercise performance parameters.

#### 4.2.1. Chronic l-C Supplementation Protocol for Moderate-Intensity Exercise Performance (50–79% $\dot{V}$O_2 max_)

More than 95% of the body’s carnitine pool is confined to skeletal muscle, where it fulfils two major metabolic roles. For any given tissue the normal carnitine content is that which is necessary for an optimal rate of long-chain fatty-acid oxidation [52]. Firstly, in mitochondrial fatty-acid translocation and transport, carnitine is a substrate for carnitine palmitoyl-transferase 1 (CPT1) [53]. For that reason, oral carnitine feeding has been advocated as a possible ergogenic aid, with the main premise that it increases muscle carnitine content. In consequence, it would increase muscle fat transport and oxidation, as well as delay muscle glycogen depletion [23,53]. In three of the studies included in this systematic review, supplementation doses of 1 to 3 g of l-C or GPL-C over a period of 4, 8, or 24 weeks did not observe significant differences in moderate-intensity exercise performance parameters [23,30,34]. In the same line, a lack of improvements in a scale of perceived exertion, $\dot{V}$O_2 peak_, and total time in tests of gradual exercise on a treadmill and 90 min on a cycle ergometer at 65% $\dot{V}$O_2 max_ was observed two 2 of the studies [30,34]. One potential reason could the inability to increase muscle carnitine concentrations, although, in the case of Broad et al. [30], no biopsies were performed to report this statement. Moreover, this suggests that improvements in moderate-intensity exercise parameters could have been found if the studies were carried out using less trained participants and if lower exercise intensities were evaluated. Therefore, variability in the type of population analyzed in the studies represents another limitation in order to find conclusive results.

On the other hand, Wall et al. [23] supplemented l-C together with the intake of CHO and found no improvements in a scale of perceived exertion during moderate-intensity exercise, but a more integrating variable is needed to argument the changes, since the perception was not enough despite the placebo effect. However, as mentioned previously, the saving of muscle glycogen and the increase in fat oxidation revealed by muscle biopsies during this test allowed an improvement in the work capacity of later tests at greater intensity [23]. Unlike Wall et al. [23], previous studies used higher doses of l-C (5 to 6 g) for shorter periods (5, 7 and 14 days) and without taking CHO, but they did not report changes in the substrate utilization during moderate-intensity exercises [18] or in the muscle carnitine content by biopsies [54].

On the contrary, the observation that l-C does not affect muscle carnitine content without prior insulin stimulation seems to be in contrast with two previous studies that used supplementation of 2 g of l-C together with CHO intake for 4 weeks in moderate- to low-intensity tests [54,55]. While muscle biopsies performed in one of them reported increases in the pyruvate dehydrogenase activity and VO_2 max_ of the participants [54], in the other one, no biopsies were carried out; however, a lower respiratory quotient was reported, suggesting that performance improvements derived from increased fat oxidation and probable muscle glycogen sparing could be expected [55]. However, in this study, there were no changes in parameters such as O_2_. Furthermore, other measurements such as oxygen uptake, heart rate, blood glycerol, and resting plasma free fatty-acid concentrations would have been necessary to report the improvement in different performance parameters [55]. For this reason, more studies are needed to be able to affirm with conviction the improvements in low–moderate-intensity performance derived from chronic supplementation protocols.

#### 4.2.2. Acute Supplementation Protocol for Moderate-Intensity Exercise Performance (50–79% $\dot{V}$O_2 max_)

Along with the intake of CHO, doses of 3 g of l-C 3 h before exercise at 65% $\dot{V}$O_2 max_ on a cycle ergometer and doses of 2 g taken 2 h before a marathon race and 20 km after start did not find significant improvements in the power achieved during 40 min on the cycle ergometer or in total time in a marathon [20,24]. In another study with l-C and l-tartrate, they used the same supplementation protocol as Burrus et al. [24] (3 g of l-C 3 h before), and no changes were found in the substrate used during a 60 min test at 60% $\dot{V}$O_2 max_ [56]. The lack of improvements in these types of studies concluded that the supplementation period was not sufficient to increase the l-C reserves in the muscle and, therefore, the substrate utilization was not modified [20,24,56]. More research is necessary to explain the results and provide greater clarity and scientific evidence about the acute supplementation protocol of l-C in moderate-intensity exercise performance and related outcomes.

Related to moderate-intensity exercise performance, different methodological approaches presented contradictory results after both acute and chronic supplementation protocols. Although more studies are needed to confirm the real effect of supplementation on moderate-intensity exercise performance parameters, this systematic review concluded that, while l-C supplementation with CHO and 2.72 g of l-C for a prolonged period of 24 weeks could increase muscle carnitine content, fat oxidation, and glycogen sparing during low-intensity testing, it seemingly had no effect on moderate-intensity exercise performance.

### 4.3. Limitations, Strengths, and Future Lines of Investigation

Interesting improvements were found in some parameters included in this systematic review, especially regarding high-intensity performance. However, factors limited conclusive results in other studies. Among these factors were an insufficient sample size, variability in the type of population analyzed, type of supplementation, and inclusion or non-inclusion of insulin stimulus with its intake, as well as lack of muscle biopsy data measuring the content of muscular carnitine. In contrast, the main strength of this systematic review is its description of the current state of the art analyzing different supplementation dosages and timing. On the other hand, the classification of high- and moderate-intensity exercise performance could not be fully adjusted as a function of the criteria. Likewise, this systematic review did not include recovery outcomes, which are an essential part of sports/exercise performance.

Future research lines should focus on homogenizing l-C supplementation protocols regarding the dose and timing, due to various current methods found in the current literature. On the other hand, the potential synergy among supplements with similar objectives (for example, ketones) should be studied, as well as the already evaluated effect of supplementation together with CHO. Therefore, more studies are needed that meet ideal conditions and that can be replicated to recommend this supplement for improving sports performance.

## 5. Conclusions

In this systematic review, evidence revealed that both acute and chronic l-C supplementation could have positive effects on high-intensity (≥80% $\dot{V}$O_2 max_) performance exercise outcomes. In particular, doses of 3 to 4 g of l-C or GPL-C ingested between 60 and 90 min before exercise could improve the lactate threshold and achieve lower levels of perceived exertion during incremental tests until exhaustion, as well as increasing peak and average power in the Wingate cycle ergometer test. Likewise, doses of 2 to 2.72 g/day of l-C for longer periods up to 9 to 24 weeks also resulted in lower blood lactate accumulation and scale of perceived exertion, along with increases in work capacity in “all out” tests, peak power in a Wingate test, and the number of repetitions and volume lifted in leg press exercises.

On the other hand, supplementation with 2.72 to 3 g/day of l-C with or without CHO intake for 4 to 24 weeks did not present improvements in performance parameters during moderate-intensity exercise (50–79% $\dot{V}$O_2 max_). Likewise, doses of 2 to 3 g of l-C ingested between 30 min to 3 h before exercise do not appear to be effective when it comes to improvements in moderate-intensity exercise performance.

## Figures and Tables

**Figure 1 nutrients-13-04359-f001:**
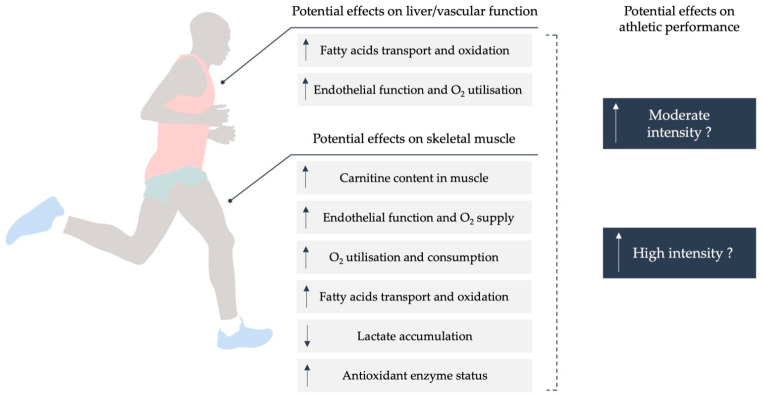
Potential effects of l-Carnitine supplementation on different physiological and metabolic pathways that could improve moderate- and high-intensity exercise performance.

**Figure 2 nutrients-13-04359-f002:**
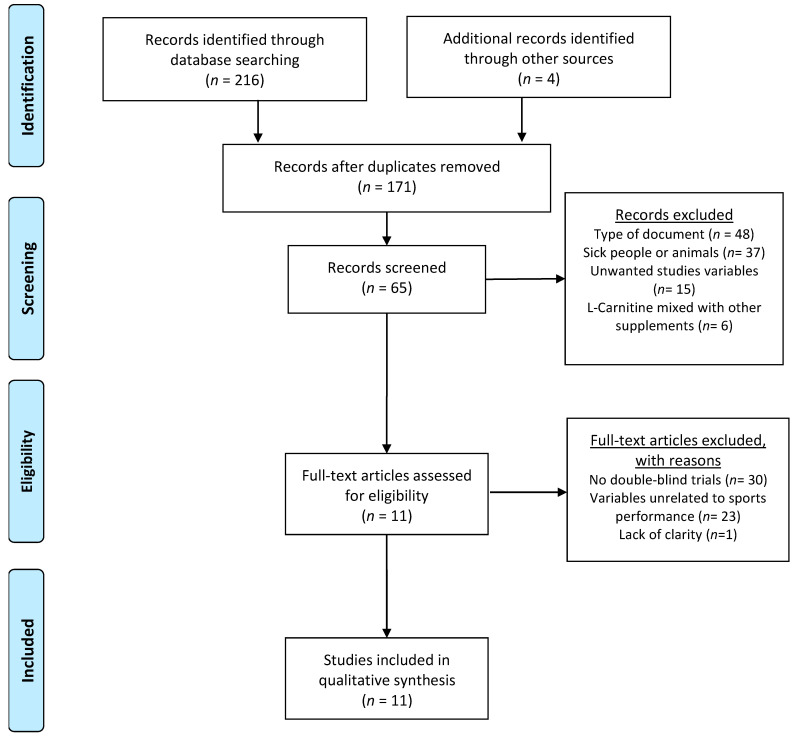
Selection of the studies included in the systematic review.

**Table 1 nutrients-13-04359-t001:** Methodological quality of the studies included in the review by McMaster Critical Review Form for Quantitative Studies.

Studies Included	ITEMS																T	%	MQ
	1	2	3	4	5	6	7	8	9	10	11	12	13	14	15	16			
Trappe et al., 1994 [14]	1	1	1	1	1	0	0	0	1	1	1	0	0	1	0	0	9	60.00	A
Colombani et al., (1996) [20]	1	1	1	1	0	1	0	1	1	0	1	0	0	1	0	0	9	60.00	A
Broad et al., (2005) [30]	1	1	1	1	0	1	0	1	1	0	1	1	1	1	0	1	12	75.00	G
Smith et al., (2008) [34]	1	1	1	1	0	1	1	1	1	1	1	1	0	1	0	1	13	86.66	VG
Jacobs et al., (2009) [32]	1	1	1	1	1	1	0	1	1	1	1	1	0	1	1	1	14	93.33	VG
Wall et al., (2011) [23]	1	1	1	1	0	1	0	0	1	1	1	1	1	1	1	1	13	81.25	VG
Orer et al., (2014) [16]	1	1	1	1	0	1	0	1	1	1	1	1	1	1	1	1	14	87.50	VG
Arazi & Mehrtash (2017) [31]	1	1	1	1	1	1	0	0	1	1	1	1	0	1	0	0	11	68.75	G
Koozehchian et al., (2018) [33]	1	1	1	1	0	1	1	0	1	1	1	1	1	1	1	1	14	87.50	VG
Burrus et al., (2018) [24]	1	1	1	1	1	1	0	1	1	0	1	1	1	1	0	1	13	81.25	VG
Shannon et al. (2018) [15]	1	1	1	0	0	1	0	0	1	0	1	1	1	1	0	1	10	62.50	A

Legend: T: (total items accomplished); 1: item accomplished; 0: item not accomplished; MQ: methodological quality; E: excellent; VG: very good; G: good; A: acceptable.

**Table 2 nutrients-13-04359-t002:** Participants and intervention characteristics of the studies included with the systematic review.

Level of participants	Professional		2 studies [16,31]
	Amateur		7 studies [14,20,23,24,30,32,33]
	Untrained		2 studies [15,34]
Type of l-Carnitine	l-Carnitine		6 studies [14,16,20,24,31,33]
	l-Carnitine L-Tartrate		3 studies [15,23,30]
	Glycine-Propionyl l-Carnitine		2 studies [32,34]
Time of administration	Chronic	4 weeks	1 study [30]
		8 weeks	1 study [34]
		9 weeks	1 study [33]
		24 weeks	2 studies [15,23]
	Acute	7 days	1 study [14]
		Before/during	5 studies [16,20,24,31,32]
Doses used	Chronic	1 g/day (Propionyl l-Carnitine)	1 study [34]
		2 g/day	2 studies [30,33]
		2,72 g/day	1 study [23]
		3 g/day (Propionyl l-Carnitine or l-Carnitine)	2 studies [15,34]
	Acute	3 g (l-Carnitine) 3 g (Propionyl l-Carnitine)	3 studies [16,24,31] 1 study [32]
		4 g (l-Carnitine)	3 studies [14,16,20]

**Table 3 nutrients-13-04359-t003:** Articles included in the systematic review that investigated the effect of l-C on high-intensity (≥80% $\dot{V}$O_2 max_) exercise performance variables.

Author/s	Population	Intervention	Outcomes Analyzed	Main Results	
Chronic l-C supplementation					
Smith et al., 2008 [34]	32 untrained participants (9 men and 23 women aged between 18 and 44 years)	1 g/day of PL-C *n* = 11 3 g/day of PL-C *n* = 12 Duration: 8 weeks	Test of anaerobic power (30 s on cycle ergometer):	GPL-C-1	GPL-C-3
			Absolute and relative peak power	↔	↔
			Absolute and relative mean power	↔	↔
			% of fatigue	↔	↔
			Absolute and relative total work	↔	↔
Wall et al., 2011 [23]	14 moderately trained recreational athletes (25.9 ± 2.1 years)	4 g/day l-CLT (2,72 g/day l-C) with 160 g of CHO distributed 2 times per day (1 at breakfast and 4 h later) Duration: 24 weeks	30 min at 80% $\dot{V}$O_2 max_ on cycle ergometer:	Week12	Week24
			Borg scale	↔	↑
			30 min “all out” on cycle ergometer:		
			Work output	↔	↑
Shannon et al., 2018 [15]	14 untrained men (23 ± 2 years)	4.50 g/day of l-CLT (3 g/day l-C) with 160 g of CHO distributed 2 times per day (1 at breakfast and 4 h later) Duration: 24 weeks	HIIT on cycle ergometer (100% $\dot{V}$O_2 max_):		
			$\dot{V}$O_2 max_	↔	
			Watt max	↔	
			Work output	↔	
Koozehchian et al., 2018 [33]	23 endurance-trained males (25 ± 2 years)	2 g/day of l-C caps of 1 g (one at breakfast and the other at lunch) Duration: 9 weeks	Performance on bench press:		
			Number of repetitions	↔	
			Third set lifting volume	↔	
			Performance on leg press:		
			Number of repetitions	↑	
			Third set lifting volume	↑	
			Anaerobic performance (30 s Wingate test):		
			Mean power and relative peak power	↔	
			Peak power and absolute peak power	↑	
Acute l-C supplementation					
Trappe et al., 1994 [14]	20 trained collegiate male swimmers (20.1 ± 0.6 years)	4 g/day of l-C in a citrus drink twice daily for 7 days	Five 91.4 m (100 yd) repeated swims at a supramaximal intensity with 2 min rest between each		
			Swimming velocity	↔	
Jacobs et al., 2009 [32]	24 resistance-trained men (25.2 ± 3.6 years)	4,5 g GPL-C (3 g PL-C) with 8 oz of water 90 min before the test	Wingate test (5 sprints of 10 s on cycle ergometer)		
			Peak Power on 3, 4, and 5 sprints	↑	
			Mean Power on 4 and 5 sprints	↑	
			Relative total mean and peak power	↔	
Orer & Guzel, 2014 [16]	26 male professional soccer players (18.42 ± 0.50)	3 g of l-C with a glass of juice (*n* = 12) LK3 4 g of l-C with a glass of juice (*n* = 14) LK4 1 h before the test	Graded exercise test on the treadmill until exhaustion:	LK3	LK4
			Speed at:		
			2, 2.5, and 3 mmol/L of blood lactate	↑	↔
			3.5 mmol/L of blood lactate	↑	↑
			4 mmol/L of blood lactate	↑	↑
			Borg Scale at:		
			8, 11, and 12 km/h	↑	↔
			13 and 14 km/h	↑	↑
			15 and 16 km/h	↔	↑
Arazi and Mehrtash (2017) [31]	18 male artistic gymnasts (21 ± 2.12 years)	3 g of l-C 90 min before the test	Anaerobic sprint test (RAST)		
			Maximum power	↑	
			Mean power	↔	
			20 m shuttle run		
			$\dot{V}$O_2 max_	↑	
Burrus et al., 2018 [24]	10 moderately active men (27 ± 4 years)	3 g l-C in 200 mL of H_2_O 3 h before the exercise 94 g of CHO in 500 mL H_2_O 2 h before the exercise 94 g of CHO in 500 mL H_2_O 30 min before the exercise	Exercise on cycle ergometer at 85% $\dot{V}$O_2 Peak_:		
			Power output	↔	
			Time to exhaustion	↔	

Legend: ↔ no difference in effects between groups (*p* > 0.05); ↑ statistically higher effects in l-C (*p* < 0.05); CHO: carbohydrates; HIIT: high-intensity interval training; l-CLT: l-Carnitine l-tartrate; l-C: l-carnitine; PL-C: propionyl l-carnitine.

**Table 4 nutrients-13-04359-t004:** Articles included in the systematic review that investigated the effect of l-Carnitine on low–moderate-intensity (50–79% $\dot{V}$O_2 max_) athletic performance variables.

Author/s	Population	Intervention	Outcomes Analyzed	Main Results	
Chronic l-C supplementation					
Broad et al., 2005 [30]	15 endurance-trained men (20 to 46 years)	3 g of l-CLT (2 g l-C) Duration: 4 weeks and 2 weeks washout period between them	90 min at 65% $\dot{V}$O_2 max_ on cycle ergometer:		
			Borg scale	↔	
			$\dot{V}$O_2_	↔	
			20 km time trial duration: Total time trial		
			Total time trial	↔	
Smith et al., 2008 [34]	32 untrained participants (9 men and 23 women (18 to 44 years)	1 g/day of PL-C *n* = 11 3 g/day of PL-C *n*= 12 Duration: 8 weeks	Aerobic exercise test on the treadmill:	GPL-C-1	GPL-C-3
			Borg scale	↔	↔
			$\dot{V}$O_2 peak_	↔	↔
			Total exercise time	↔	↔
Wall et al., 2011 [23]	14 Moderately trained recreational athletes (25.9 ± 2.1 years)	4 g of l-CLT (2,72 l-C) with 160 g of CHO distributed 2 times per day (1 at breakfast and 4 h later Duration: 24 weeks	Test on cycle ergometer at 50% $\dot{V}$O_2 max_:	week 12	week 24
			Borg scale	↔	↔
Acute l-C supplementation					
Colombani et al., 1996 [20]	7 endurance-trained men (36 ± 3 years)	2 g of l-C 2 h before the marathon and 2 g of l-C after 20 km 125 mL of sugar tea every 5 km and 500 mL during the next hour after the marathon	Marathon:		
			Total time	↔	
Burrus et al., 2018 [24]	10 moderately active men (27 ± 4 years)	3 g of l-C + 200 mL of H_2_O 3 h before the exercise 94 g of CHO in 500 mL of H_2_O 2 h before the exercise 94 g of CHO in 500 mL of H_2_O 30 min before the exercise	Test on cycle ergometer at 65% $\dot{V}$O_2 peak_:		
			Power at 0, 10, 20, 30 y 40 min	↔	

Legend: ↔ no difference in effects between groups (*p* > 0.05); CHO: carbohydrates; l-CLT: L-Carnitine l-tartrate; l-C: l-carnitine; PL-C: propionyl l-carnitine.

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
