# Peer review of "Effect of Acute and Chronic Oral l-Carnitine Supplementation on Exercise Performance Based on the Exercise Intensity: A Systematic Review"

_nutrients, 2021, doi:10.3390/nu13124359_

Round 1

Reviewer 1 Report

“Effect of acute and chronic L-Carnitine supplementation on anaerobic and aerobic sport performance: a systematic review” 

Overall: This is a good systematic review and is needed to clarify the current state of knowledge about L-carnitine supplementation in sports performance. The authors presented a reasonable justification for this study, good methodological approach and results, and and a reasonable attempt in discussion.

Overall, the manuscript is well organised and nicely written. However, there is a potential for it to be significantly improved following a major revision.

Please address each comment in a point by point response:

Generic comments:

-The  background could benefit from a proper scoping of further physiological mechanisms, beyond sport science studies ( clarify the multipurpose nutritional benefits and mechanisms of such supplement and then link them to mechanisms related to optimising performance).

-The justification based on fat-loss mechanisms (which needs expanding on here) may not be sufficient to decide on whether a supplement is useful for sports performance. There needs to be a clear distinction between “fat-loss” mechanisms and “fat-metabolism” mechanisms and how they relate to affecting performance in the acute and chronic phase. Also, other mechanisms could also play a role here (e.g. oxidative stress and muscle damage). Some argue that anti-oxidation mechanisms may counteract ergogenic sports performance mechanisms, and L-carnitine may fall into this category. This makes it important to critically compare/critique methodologies/protocols of studies and debate different points of view. Perhaps consider this point throughout the background and discussion.

-Define a clear criteria for what is considered "sport performance" for each "aerobic" and "anaerobic". -Consider a different criteria to divide your results because there seems to be a mix-up between "aerobic" and "anaerobic" findings analysed (see comments below). 

-The terminology “aerobic” vs. “anaerobic” makes it important to clearly define your performance criteria under those two terms. However, this may be difficult where some studies include elements of both (e.g. ref. 5). Also 80% or 85% of VO2max is aerobic. You might consider criteria based on “exercise intensity” instead of aerobic/anaerobic.

Specific comments:

Abstract:

-Lines 16-20 are providing some repetitions in the background and aims (e.g. effects on aerobic and anaerobic ..). You can summarise those into 1-2 lines max in order to use the space to provide more details in results. Perhaps start with “The aim of..”

-Please correct the statement: “in both ways high intensity and aerobic sports disciplines” because “high intensity” can also be aerobic. Perhaps use “anaerobic or aerobic types of sports”

-Use “tests” (plural for both aerobic and anaerobic), also adjust the statement “The effects of acute and chronic supplementation on both…..tests were measured”

-Use “and 5 studies”

-“ Showing improvement” in what?

-Lines 30-36: Provide some details about timing/dose/type of improvement or performance/settings (or examples of most important findings). You may also need to distinguish between aerobic and anaerobic effects acutely and chronically.

Introduction:

-Line 51: elaborate on the metabolic mechanisms in which aerobic performance could be enhanced through expansion of carnitine pool, and whether there are other points of views on such link.

-Line 63: Please specify reference 10 (e.g. page number in reference list).

-Line 67: Specify or rephrase “in this area”  

-Line 72: Rephrase “determinant parameters such as …” you mean “sport performance determinants such as..”?

-Line 72-73: The statement needs rephrasing it is confusing. Specify what type of sport performance? If you use VO2max you need to link it with anaerobic-type performance because it can also be used as a determinant for aerobic performance. There are also several other indicators of “anaerobic sports performance” which you used correctly “e.g. speed, sprint, power output, fatigue index, Rep max..etc.).

-Line 73-79: it’d be good to see some details about the magnitude of increase (in Watt, % or relevant units depending in the studies found).

-Line 93: No change “was” or no changes “were”. Check English throughout the manuscript please.

-Page 2: Paragraphs 1-3: You present good studies but you need to make more critical comparisons to make the argument stronger about the need for such systematic review. Specific details, contrasting points of view and details about the topic reviewed (timing, its interaction between dose and acute/chronic mechanisms of effects). Perhaps add specific statements within those paragraphs or refine those paragraphs and add another paragraph to make the case stronger.

Methods:

-Very good section and sub-headings overall.

-Line 111: Just to double-check, are you sure you have only included only “double blind”, even when looking at the chronic effects in training studies?

-Line 139: Check the Spanish word and correct “analysis”

-Paragraph 2.1: Did your inclusion criteria consider the L-C effects on sport or exercise recovery? Some consider it an essential part of sports/exercise performance (Strength type, repeated exercise bouts.. etc.). Please provide a clear justification.

Results:

-Line 228-230: Please double-check whether your selected definition of “sport performance variables” may have missed any term, which result in an exclusion of any useful study.

-Table 3: References 18: Some would have thought that a Marathon (running at 4 mM of BLC) would be “aerobic”. The same goes for reference 21 and part of 5 (80% or 85%VO2).  Please check the whole manuscript for consistent terminology.

-Table 4: Insert columns definition (as you did in Table 3).

Discussion:

-Lines 371-377; Section 4.1: I do not see how this section is describing “anaerobic”. It is discussing fat metabolism and its intensity-dependent changes. Glycogen sparing mechanisms are not anaerobic. Authors need to check the whole manuscript because the terminology “anaerobic/aerobic” is confused in this manuscript which is a major point (see my suggestion above).

-383-388: The same issue as above. This is describing intensity-dependent mechanisms, which are primarily aerobic.

-Line 420 (whole paragraph): HIT is not necessarily anaerobic. It is actually aerobic.

-Line 442-445: Lactate thresholds related effects (reference 4) are for an “endurance aerobic performance”. Please see comments above.

-578-593: Limitation section should be more specific and based on actual limitation of this this systematic review, rather than a future recommendation.

Author Response

Point-by-Point Response to Reviewer’s Comments

We would like to sincerely thank the reviewers for their helpful recommendations. We have seriously considered all the comments and carefully revised the manuscript accordingly. Revisions are highlighted in yellow through the manuscript to indicate where changes have taken place. We feel that the quality of the manuscript has been significantly improved with these modifications and improvements based on the reviewers’ suggestions and comments. We hope our revision will lead to an acceptance of our manuscript for publication in Nutrients.

In advance,

Kind regards

REVIEWER 1

Suggestions for Authors

“Effect of acute and chronic L-Carnitine supplementation on anaerobic and aerobic sport performance: a systematic review” 

Overall: This is a good systematic review and is needed to clarify the current state of knowledge about L-carnitine supplementation in sports performance. The authors presented a reasonable justification for this study, good methodological approach and results, and and a reasonable attempt in discussion.

Overall, the manuscript is well organised and nicely written. However, there is a potential for it to be significantly improved following a major revision.

Please address each comment in a point by point response:

Generic comments:

REVIEWER: -The background could benefit from a proper scoping of further physiological mechanisms, beyond sport science studies (clarify the multipurpose nutritional benefits and mechanisms of such supplement and then link them to mechanisms related to optimising performance).

AUTHORS: Thank you for your recommendation. The authors have made a figure which detail some possible mechanism that the carnitine supplementation could be effective on anaerobic and aerobic performance.

REVIEWER: -The justification based on fat-loss mechanisms (which needs expanding on here) may not be sufficient to decide on whether a supplement is useful for sports performance. There needs to be a clear distinction between “fat-loss” mechanisms and “fat-metabolism” mechanisms and how they relate to affecting performance in the acute and chronic phase.

AUTHORS: Thank you for your interest. The authors have included some mechanism where the carnitine could act improving body composition: “improving insulin resistance and may decrease appetite and food intake through a direct effect on hypothalamus”.

REVIEWER:  Also, other mechanisms could also play a role here (e.g. oxidative stress and muscle damage). Some argue that anti-oxidation mechanisms may counteract ergogenic sports performance mechanisms, and L-carnitine may fall into this category. This makes it important to critically compare/critique methodologies/protocols of studies and debate different points of view. Perhaps consider this point throughout the background and discussion.

AUTHORS: Thank you for your recommendation. The authors have included the oxidative stress effect: “[9]. Likewise, the antioxidant effect of carnitine by increased overall antioxidant enzyme status, could be effective on muscle recovery [10].”

REVIEWER: -Define a clear criteria for what is considered "sport performance" for each "aerobic" and "anaerobic".

AUTHORS: Thank you for your recommendation. Based on your below recommendation the terms anaerobic and aerobic have been changed by high (≥80% VO2 max) and low-moderate intensity (<80% VO2 max) performance.

REVIEWER: -Consider a different criteria to divide your results because there seems to be a mix-up between "aerobic" and "anaerobic" findings analysed (see comments below). 

AUTHORS: Thank you for your comment. Following previous response, we have changed that division.

REVIEWER: -The terminology “aerobic” vs. “anaerobic” makes it important to clearly define your performance criteria under those two terms. However, this may be difficult where some studies include elements of both (e.g. ref. 5). Also 80% or 85% of VO2max is aerobic. You might consider criteria based on “exercise intensity” instead of aerobic/anaerobic.

AUTHORS: Thank you for your recommendation. The authors have changed the criteria based on the exercise intensity. High (≥80% VO2 max) and low-moderate intensity (<80% VO2 max) performance

Specific comments:

Abstract:

REVIEWER: -Lines 16-20 are providing some repetitions in the background and aims (e.g. effects on aerobic and anaerobic ..). You can summarise those into 1-2 lines max in order to use the space to provide more details in results. Perhaps start with “The aim of..”

AUTHORS: Thank you for your recommendation. The authors have rewritten these lines: “L-Carnitine (L-C) and any of its forms (Glycin-Propionyl L-Carnitine (GPL-C) or L-Carnitine L-tartrate (L-CLT)) has been frequently recommended as a supplement to improve sports performance due to among others its role in fat metabolism and in maintaining the mitochondrial Acetyl CoA/CoA ratio. The main aim of the present systematic review was to determine the effects of oral L-C supplementation on exercise performance in high (≥80% VO2 max) and low-moderate (<80% VO2 max) performance and to show the effective doses and ideal timing of its intake.”

REVIEWER: -Please correct the statement: “in both ways high intensity and aerobic sports disciplines” because “high intensity” can also be aerobic. Perhaps use “anaerobic or aerobic types of sports”

AUTHORS: Thank you for your recommendation. As the authors indicated in the previous comment, we have rewritten these lines.

REVIEWER: -Use “tests” (plural for both aerobic and anaerobic), also adjust the statement “The effects of acute and chronic supplementation on both…..tests were measured”

AUTHORS: Thank you. Done.

REVIEWER: -Use “and 5 studies”

AUTHORS: Thank you. Done.

REVIEWER: -“ Showing improvement” in what?

AUTHORS: Thank you for your recommendation. The authors have rewritten this part of abstract: “Four of them, measured the effects of chronic supplementation (showing improvements on in lower blood lactate accumulation and scale of perceived exertion punctuation, achieving increases in work capacity in “all-out” tests, in peak power in a Wingate test and in the number of repetitions and volume lifted in leg press exercises) and 5 studies analysed the effects of acute supplementation (showing improvements on the lactate threshold and achieve lower levels of perceived exertion during incremental tests until exhaustion, as well as increasing peak and average power in the Wingate Cycle Ergometer Test).”

REVIEWER: -Lines 30-36: Provide some details about timing/dose/type of improvement or performance/settings (or examples of most important findings). You may also need to distinguish between aerobic and anaerobic effects acutely and chronically.

AUTHORS: Thank you for your recommendation. The authors have rewritten this part of abstract: “Six studies used L-C, while 3 studies used L-CLT and 2 others combined the molecule Propionyl L-Carnitine (PL-C) with GPL-C. Five studies analysed the chronic supplementation (4 - 24 weeks) and 6 studies used an acute administration (< 7 days). The administration doses in this chronic supplementation varied from 1 to 3 g/day and in acute supplementation, oral L-C supplementation doses ranged from 3 to 4 g. On the one hand, the effects of oral L-C supplementation on high in-tensity performance variables were analysed in 9 studies.”

Introduction:

REVIEWER: -Line 51: elaborate on the metabolic mechanisms in which aerobic performance could be enhanced through expansion of carnitine pool, and whether there are other points of views on such link.

AUTHORS: Thank you for your recommendation. The mechanisms which aerobic performance could be enhanced through expansion of carnitine pool are explained in the second paragraph.

REVIEWER: -Line 63: Please specify reference 10 (e.g. page number in reference list).

AUTHORS: Thank you for your recommendation. The authors have changed that reference by:

Gnoni, A.; Longo, S.; Gnoni, G. V.; Giudetti, A.M. Carnitine in human muscle bioenergetics: Can carnitine supplementation improve physical exercise? Molecules 2020, 25, doi:10.3390/molecules25010182.

REVIEWER: -Line 67: Specify or rephrase “in this area”  

AUTHORS: Thank you for your interest. The authors have changed that paragraph: “However, in order that these hypotheses can be fulfilled, it is essential to maintain a high content of muscle carnitine [12]. Nonetheless, it’s difficult to increase muscle content due to the transmembrane gradient for L-C through the skeletal muscle is high, facilitating the carnitine output from muscle to the plasma [13]. Therefore, oral L-C supplementation and any of its forms (Glycin-Propionyl L-Carnitine (GPL-C) or L-Carnitine L-tartrate (L-CLT)) could be an effective strategy in order to achieve a carnitine load in the muscle and improve both high and low-moderate in-tensity performance [14].”

REVIEWER: -Line 72: Rephrase “determinant parameters such as …” you mean “sport performance determinants such as..”?

AUTHORS: Thank you. Done.

REVIEWER: -Line 72-73: The statement needs rephrasing it is confusing. Specify what type of sport performance? If you use VO2max you need to link it with anaerobic-type performance because it can also be used as a determinant for aerobic performance. There are also several other indicators of “anaerobic sports performance” which you used correctly “e.g. speed, sprint, power output, fatigue index, Rep max..etc.).

AUTHORS: Thank you for your recommendation. The authors have included more information: “In fact, previous studies have reported that long-term oral L-C supplementation (2 g/day for 4 weeks) produced certain improvements on sport performance determinants such as VO2 max (ml/min/kg) and Maximal work output (Kj) in both elite and amateur athletes [2].”

REVIEWER: -Line 73-79: it’d be good to see some details about the magnitude of increase (in Watt, % or relevant units depending in the studies found).

AUTHORS: Thank you for your recommendation. The authors have included more information: “VO2 max (ml/min/kg) and Maximal work output (Kj)”

REVIEWER: -Line 93: No change “was” or no changes “were”. Check English throughout the manuscript please.

AUTHORS: Thank you. Done.

REVIEWER: -Page 2: Paragraphs 1-3: You present good studies but you need to make more critical comparisons to make the argument stronger about the need for such systematic review. Specific details, contrasting points of view and details about the topic reviewed (timing, its interaction between dose and acute/chronic mechanisms of effects). Perhaps add specific statements within those paragraphs or refine those paragraphs and add another paragraph to make the case stronger.

AUTHORS: Thank you for your recommendation. The introduction has been changed substantially traying to meet the recommendations of the reviewers.  

Methods:

REVIEWER: -Very good section and sub-headings overall.

AUTHORS: Thank you for your comment.

REVIEWER: -Line 111: Just to double-check, are you sure you have only included only “double blind”, even when looking at the chronic effects in training studies? -Line 139: Check the Spanish word and correct “analysis”

AUTHORS: Thank you. Done.

REVIEWER: -Paragraph 2.1: Did your inclusion criteria consider the L-C effects on sport or exercise recovery? Some consider it an essential part of sports/exercise performance (Strength type, repeated exercise bouts.. etc.). Please provide a clear justification.

AUTHORS: Thank you for your interest. We agree that recovery is essential for sport performance. However, in this systematic review only has included outcomes about high and low-moderate intensity performance. The authors have included this limitation in the limitations section: “Likewise, this systematic review has not included recovery outcomes that it is an essential part of sports/exercise performance.”

Results:

REVIEWER: -Line 228-230: Please double-check whether your selected definition of “sport performance variables” may have missed any term, which result in an exclusion of any useful study.

AUTHORS: Thank you for your interest. The authors have changes sport performance variables by outcomes related with sport performance variables.

REVIEWER: -Table 3: References 18: Some would have thought that a Marathon (running at 4 mM of BLC) would be “aerobic”. The same goes for reference 21 and part of 5 (80% or 85%VO2).  Please check the whole manuscript for consistent terminology.

AUTHORS: Thank you for your interest. As you can see the authors have changed the criteria by high and low-moderate intensity performance.

REVIEWER: -Table 4: Insert columns definition (as you did in Table 3).

AUTHORS: Thank you for your observation. The authors have added that information.

Discussion:

REVIEWER: -Lines 371-377; Section 4.1: I do not see how this section is describing “anaerobic”. It is discussing fat metabolism and its intensity-dependent changes. Glycogen sparing mechanisms are not anaerobic. Authors need to check the whole manuscript because the terminology “anaerobic/aerobic” is confused in this manuscript which is a major point (see my suggestion above).

AUTHORS: Thank you for your recommendation. As we have already mentioned earlier, the Exercise Division has been carried out based on High or low-moderate intensity.

REVIEWER: -383-388: The same issue as above. This is describing intensity-dependent mechanisms, which are primarily aerobic.

AUTHORS: Thank you for your recommendation. As we have already mentioned earlier, the Exercise Division has been carried out based on High or low-moderate intensity.

REVIEWER: -Line 420 (whole paragraph): HIT is not necessarily anaerobic. It is actually aerobic.

AUTHORS: Thank you for your recommendation. As we have already mentioned earlier, the Exercise Division has been carried out based on High or low-moderate intensity.

REVIEWER: -Line 442-445: Lactate thresholds related effects (reference 4) are for an “endurance aerobic performance”. Please see comments above.

AUTHORS: Thank you for your recommendation. As we have already mentioned earlier, the Exercise Division has been carried out based on High or low-moderate intensity.

REVIEWER: -578-593: Limitation section should be more specific and based on actual limitation of this this systematic review, rather than a future recommendation.

AUTHORS: Thank you for your recommendation. The authors have included some limitations: On the other hand, the classification of high and low-moderate intensity performance could not be fully adjusted in function of the criteria that are used for that partition. Likewise, this systematic review has not included recovery outcomes that it is an essential part of sports/exercise performance.

Reviewer 2 Report

Thank you for providing an interesting review of a supplement that's of growing interest. I think the review's main strengths are that you acknowledge the lack of evidence of an effect, despite a plausible mechanism and the methods are robust and thoughtful, leading to a well-conducted review. However, I feel it could be improved by detailing some of the possibly ergogenic mechanisms further, possibly through an illustration/figure, which would lead to a richer introduction for readers who are not familiar with L-carnitine. Please also consider some general English revisions throughout, e.g. L-carnitine would be sufficient, no 'the' is required nor an abbreviation, as no words are saved the abbreviation feels less specific. Please find below my line by line review. 46 - it's may be replaced by is 47 - can you be more specific here than improves aerobic metabolism? My understanding is that it's hypothesised that L-carnitiine increases the aerobic contribution to exercise by increasing fat oxidation? 48 - improving body composition may work better here than 'body fat loss' as it links better to the next point, which is also positive/improvement 50 - please amend to 'the muscle carnitine pool' 53 - please amend inside to in 56 - please place a comma (,) after oxidation 59 - following Moreover, you may wish to state 'L-carnitine may' or amend enhance to enhancements 62 - consider amending 'some' to 'multiple', as this is a bit more direct 65/67 - please revisit and rephrase this sentence slightly. The wording isn't quite right, but this is a key sentence for your introduction, as it nicely ties mechanisms and performance together. 78 - by Borg scales, do you mean rating of perceived exertion? If so, please amend 83 - please amend this to these; can you also expand a little more as to why no improvements in these parameters were found - was the duration or dose insufficient to increase the muscle carnitine pool? 93 - please amend change to changes 96 - you've cited plenty of studies that have shown no effect or small effects, is it that the mechanism for increasing and utilising L-carnitine makes sense, but this isn't transferring into improved performance? Or have studies that have used biopsies confirmed that an increase in the intramuscular pool is occurring, but this is not transferring to performance with/without CHO intake? 153 - great to see a range of L-carnitine supplements being used. I'd recommend including some comment upon different types either in the introduction (or discussion, I've not got there yet) 156 - when you say 'on sports performance' here, does that mean a particular aspect of performance related to that sport? Or an actual game or event outcome? Please reword to clarify this 178 - I've never heard of the snowball strategy before; interesting to see that this approach to systematic reviews has a name, as opposed to just being best practice There seems to be some slight disagreement between your inclusion criteria and outcome measures section, please revise so that they reflect each other and the process undertaken more clearly 226/227 - you state both 55 and 54 when talking about the number of articles included for full-text review. Please revise accordingly. 268 - please amend Firstable to Firstly 289 - I don't think Finally is necessary here, you could perhaps add 'A further six studies...' The section 3.3.1 combines discussion and results style reporting of studies. It may be best to keep this simpler here, as you have done in 3.3.2, and condense some text so that findings can be more accurately discussed in the discussion, whilst still referring to the appropriate Table(s) Given the volume of changes provided above, I would prefer to see them implemented and then review a revised version of the paper before taking a look at the discussion. This is to balance the authors' workload, as I'm aware that I have already requested quite considerable revision, further detail and the production of a Figure. Having said that, a quick scan of the paper suggests that intravenous l carnitine injections are not mentioned at all? I find this curious, as this method of administration clearly supports the proposed mechanisms, especially in clinical settings e.g. dialysis. It may be important to mention this, as this practice whilst effective would currently be banned in sport.

Author Response

Point-by-Point Response to Reviewer’s Comments

We would like to sincerely thank the reviewers for their helpful recommendations. We have seriously considered all the comments and carefully revised the manuscript accordingly. Revisions are highlighted in yellow through the manuscript to indicate where changes have taken place. We feel that the quality of the manuscript has been significantly improved with these modifications and improvements based on the reviewers’ suggestions and comments. We hope our revision will lead to an acceptance of our manuscript for publication in Nutrients.

In advance,

Kind regards

REVIEWER 2

Thank you for providing an interesting review of a supplement that's of growing interest. I think the review's main strengths are that you acknowledge the lack of evidence of an effect, despite a plausible mechanism and the methods are robust and thoughtful, leading to a well-conducted review.

REVIEWER: However, I feel it could be improved by detailing some of the possibly ergogenic mechanisms further, possibly through an illustration/figure, which would lead to a richer introduction for readers who are not familiar with L-carnitine.

AUTHORS: Thank you for your recommendation. The authors have included one figure summarized the potential effects on liver/vascular function, and on skeletal muscle.

Figure 1. Potential effects of L-carnitine supplementation on different physiological and metabolic pathways that could improve both high (≥80 VO2max) and low-moderate intensity (<80 VO2max) performance.

REVIEWER: Please also consider some general English revisions throughout, e.g. L-carnitine would be sufficient, no 'the' is required nor an abbreviation, as no words are saved the abbreviation feels less specific. Please find below my line by line review.

AUTHORS: Thank you. Done.

REVIEWER: 46 - it's may be replaced by is

AUTHORS: Thank you. Done.

REVIEWER: 47 - can you be more specific here than improves aerobic metabolism? My understanding is that it's hypothesised that L-carnitiine increases the aerobic contribution to exercise by increasing fat oxidation?

AUTHORS: Thank you. Done.

REVIEWER: 48 - improving body composition may work better here than 'body fat loss' as it links better to the next point, which is also positive/improvement

AUTHORS:

REVIEWER: 50 - please amend to 'the muscle carnitine pool'

AUTHORS: Thank you. Done.

REVIEWER: 53 - please amend inside to in

AUTHORS: Thank you. Done.

REVIEWER: 56 - please place a comma (,) after oxidation

AUTHORS: Thank you. Done.

REVIEWER: 59 - following Moreover, you may wish to state 'L-carnitine may' or amend enhance to enhancements

AUTHORS: Thank you. Done.

REVIEWER: 62 - consider amending 'some' to 'multiple', as this is a bit more direct

AUTHORS: Thank you. Done.

REVIEWER: 65/67 - please revisit and rephrase this sentence slightly. The wording isn't quite right, but this is a key sentence for your introduction, as it nicely ties mechanisms and performance together.

AUTHORS: Thank you for your recommendation. The authors have rephrased that sentence: “However, in order to fulfil these hypotheses, it is essential to maintain a high content of muscle carnitine [11]. Nonetheless, it’s difficult to increase muscle content due to elevated transmembrane gradient for L-C through the skeletal muscle, which facilitates carnitine output from muscle to the plasma [12]. Therefore, L-C oral supplementation or another supplement with L-C (Glycin-Propionyl L-Carnitine (GPL-C) or L-Carnitine L-tartrate (L-CLT)) could be an effective strategy to achieve an increase in carnitine content in the muscle [13].”

REVIEWER: 78 - by Borg scales, do you mean rating of perceived exertion? If so, please amend

AUTHORS: Thank you. Done.

REVIEWER:83 - please amend this to these; can you also expand a little more as to why no improvements in these parameters were found - was the duration or dose insufficient to increase the muscle carnitine pool?

AUTHORS: Thank you for your interest. The authors have added more information in that paragraph: “On the other hand, the first oral L-C supplementation studies failed to obtain improvements on low intensity performance, measured as changes in heart rate responses during different cycle tests at 50% of VO2 max after supplementation with 1 g/day for 14 or 28 days [16] or 5 g/day for 5 days [17]. Likewise, Soop et al., did not show differences in substrates oxidation or blood lactate concentration during prolonged low-intensity cycling exercise [17]. These results may have been because of duration and/or dose was insufficient to increase the muscle carnitine pool and fatty acid oxidation during prolonged exercise [18,19]. However, Stephens et al., demonstrated with muscle biopsies that an insulin stimulus, by using insulin intravenous infusions [20,21] or CHO intake [11,12], increased carnitine of muscle content after oral L-C supplementation”

REVIEWER: 93 - please amend change to changes

AUTHORS: Thank you. Done.

REVIEWER: 96 - you've cited plenty of studies that have shown no effect or small effects, is it that the mechanism for increasing and utilising L-carnitine makes sense, but this isn't transferring into improved performance? Or have studies that have used biopsies confirmed that an increase in the intramuscular pool is occurring, but this is not transferring to performance with/without CHO intake?

AUTHORS: Thank you for your interest. The results in this sense are controversial. While some articles show an increase in carnitine muscle content with better sport performance in other studies this carnitine content seems no sufficient to produce significant exercise improvements. Therefore, the authors have added this sentence to justify this controversial data: “These results could indicate that in addition to the content of muscle carnitine pool, there could be other mechanisms by which L-C could be effective in sports performance.”

REVIEWER: 153 - great to see a range of L-carnitine supplements being used. I'd recommend including some comment upon different types either in the introduction (or discussion, I've not got there yet)

AUTHORS: Thank you for your recommendation. The authors have included this sentence to name the other forms of obtain carnitine : “Therefore, oral L-C supplementation and any of its forms (Glycin-Propionyl L-Carnitine (GPL-C) or L-Carnitine L-tartrate (L-CLT)) could be an effective strategy in order to achieve a carnitine load in the muscle and improve both aerobic and anaerobic performance [13]. Further studies should focus on studying differences between L-C supplements.”

Moreover, the authors have included these forms of L-C supplementation (GPL-C and L-CLT) in the objective.

REVIEWER: 156 - when you say 'on sports performance' here, does that mean a particular aspect of performance related to that sport? Or an actual game or event outcome? Please reword to clarify this

AUTHORS: Thank you for your comment. The authors have changed some details about sports performance: “at least one of the variables measured was on high (≥ 80% VO2max) and low-moderate (< 80% VO2max)”

REVIEWER: 178 - I've never heard of the snowball strategy before; interesting to see that this approach to systematic reviews has a name, as opposed to just being best practice There seems to be some slight disagreement between your inclusion criteria and outcome measures section, please revise so that they reflect each other and the process undertaken more clearly

AUTHORS: “The scientific literature was recollected related to the effects of oral L-C, GPL-C or L-CLT supplementation on both high (≥80% VO2 max) and low-moderate (<80% VO2 max) intensity performance and second to share the potential effective doses and ideal timing of their intake.”

REVIEWER: 226/227 - you state both 55 and 54 when talking about the number of articles included for full-text review. Please revise accordingly.

AUTHORS: Thank you for your observation. The authors have changed 55 and 55 by 65.

REVIEWER: 268 - please amend Firstable to Firstly

AUTHORS: Thank you. Done.

REVIEWER: 289 - I don't think Finally is necessary here, you could perhaps add 'A further six studies...'

AUTHORS: Thank you. Done.

REVIEWER: The section 3.3.1 combines discussion and results style reporting of studies. It may be best to keep this simpler here, as you have done in 3.3.2, and condense some text so that findings can be more accurately discussed in the discussion, whilst still referring to the appropriate Table(s)

AUTHORS: Thanks for your recommendation. After reviewing Section 3.3.1, authors only comment, without discussing, the results of both acute supplementation and chronic L-C in anaerobic performance. The main differences with Section 3.3.2 are that 10 studies are included in 3.3.1 section and in 3.3.2 section are only 4 studies. This fact makes section 3.3.2 be significantly shorter.

REVIEWER: Given the volume of changes provided above, I would prefer to see them implemented and then review a revised version of the paper before taking a look at the discussion. This is to balance the authors' workload, as I'm aware that I have already requested quite considerable revision, further detail and the production of a Figure. Having said that, a quick scan of the paper suggests that intravenous l carnitine injections are not mentioned at all? I find this curious, as this method of administration clearly supports the proposed mechanisms, especially in clinical settings e.g. dialysis. It may be important to mention this, as this practice whilst effective would currently be

AUTHORS: Thank you for this first review. After making this first round of revisions, the authors expect the item to improve ostensibly.

Round 2

Reviewer 1 Report

Round 2:

I’d like to thank the authors for addressing my comments. It reads much better. However, I still think that a further round of revision is needed to address a couple of major points. I am happy to have another look once these have been addressed sufficiently and carefully.

Please see below.

Generic:

1-The division based on intensity is good. However, the authors need to go through some paragraphs to make sure the literature is discussed correctly in this context (not simply replacement). Where anaerobic power is clearly measured, then authors need to refer to those accurately. This is not a matter of changing the terminology, it is a crucial part of understanding the topic discussed here about sports performance determinants and potential carnitine effects found on specific performance parameters. I give a couple of examples where such issue makes the reader confused. Paragraph 4.1 in discussion is referring to a primarily anaerobic performance, since it did measure relevant “all-out” performance of peak and mean power, and also measured “blood lactate concentration” post Wingate Test (please also correct the statement in line 421 that lactate is measured “post” not “during”).

2- The authors state that running a Marathon at 4mmol/l is low to moderate intensity (Discussion paragraph 4.2) but this is a clearly heavy intensity not a low or moderate intensity. The current division require a clear justification based on exercise performance (e.g. authors could refer to weight-loss intensities common to be at low or moderate, clinical population training intensities also common to be at low to moderate). However, the studies selected here need to be clearly in the “low to moderate”. Otherwise, authors may consider introducing a 3rd division so they have “low to moderate” “moderate to heavy” and “severe intensity”. I suggest authors to check relevant literature to help authors appreciate such distinction between intensities in a sporting or clinical contexts (keywords: Predictors of exercise performance; Exercise intensity for health and sports performance, oxygen uptake kinetics). Another suggestion is that you consider two divisions (Below HIIT and HIIT), then you can define those divisions based on specific and consistent criteria.

3-As mentioned in the previous round, authors need to distinguish between body composition “fat loss” outcomes (e.g. weight or fat loss, changes in body fat percentage, increase in muscle mass. etc.) which are very different and “fat metabolism” that the authors have already discussed throughout. Reference 4 did not measure or tested the carnitine effects on body composition. I suggest removing the statement referring to carnitine effects on “body composition” since no evidence was provided here on such effects, or make relevant amendments.

If the authors wish to add this effect (perhaps in Discussion or in Limitation sections), then it needs to be correctly discussed, and there are others who have discussed this in relation to sports performance e.g. https://clinicalnutritionespen.com/article/S2405-4577(20)30053-X/fulltext) or (https://jissn.biomedcentral.com/articles/10.1186/s12970-020-00377-2).

Specific

-Double-check spellings and grammar throughout: e.g. Line 57 “increases” should be “increasing”. Line 56 “and in addition to” should remove “in addition to” or remove “and”. Line 58 “improving body composition, improving insulin resistance and may decrease…” should be “improving body composition and insulin resistance, and decreasing” Line 551 “systematic”.etc.

Line 60 sentence too long, remove “since” and start new sentence: “Previous evidence has shown that carnitine supplementation…”

Line 75: “potential multiple effects..”

Line 77: Needs rephrasing and to be more specific, e.g. “these mechanisms have been suggested/hypothesised to impact sports performance through increasing and maintaining a high muscle carnitine content”

-Reference 15: My previous comment, about adding reference page number or chapter, still needs addressing since you moved what was reference 10 and made it now reference 15.

-Line 174: Were “double-blind” studies only included?

-Line 436: Specify which type of performance. Please refer to my comment above for accuracy. Reference 17 and 30 are different from 31.

-Discussion paragraph 4.2: See comment above about distinctions between low, moderate, heavy, severe and all-out intensities.

-VO2/VO2max: Insert a Dot above the” V”.  

-Line 644: Remove “Patents” , just keep it as “Author contribution”

Author Response

Point-by-Point Response to Reviewer’s Comments

We would like to sincerely thank again to the reviewer for his/her helpful recommendations. We have seriously considered all the comments and carefully revised the manuscript accordingly. Revisions are highlighted in green through the manuscript to indicate where changes have taken place. We feel that the quality of the manuscript has been significantly improved with these modifications and improvements based on the reviewers’ suggestions and comments. We hope our revision will lead to an acceptance of our manuscript for publication in Nutrients.

In advance,

Kind regards

REVIEWER 1

REVIEWER: 1-The division based on intensity is good. However, the authors need to go through some paragraphs to make sure the literature is discussed correctly in this context (not simply replacement). Where anaerobic power is clearly measured, then authors need to refer to those accurately. This is not a matter of changing the terminology, it is a crucial part of understanding the topic discussed here about sports performance determinants and potential carnitine effects found on specific performance parameters. I give a couple of examples where such issue makes the reader confused. Paragraph 4.1 in discussion is referring to a primarily anaerobic performance, since it did measure relevant “all-out” performance of peak and mean power, and also measured “blood lactate concentration” post Wingate Test (please also correct the statement in line 421 that lactate is measured “post” not “during”).

AUTHORS: Thank you for your recommendation. The authors have modified and included more information in 4.1 and 4.2 sections about, the potential carnitine effects on high and moderate intensity performance:

In this sense, during high intensity exercise muscle carnitine loading by L-C supplementation could result in a better matching of glycolytic, pyruvate dehydrogenase complex (PDC) and mitochondrial flux, thereby reducing muscle anaerobic energy generation [23]. In this sense, carnitine's principal physiological roles during high-intensity exercise goes from acetyl group buffering (i.e. generating acetylcarnitine) to maintaining a pool of free co-enzyme A, which is required for mitochondrial flow to proceed (including the pyruvate dehydrogenase complex reaction) [38]. However, there is still a rise in the acetyl-coenzyme A (acetyl-CoA)/CoASH ratio during this type of activity, presumably due to the considerable depletion of the free carnitine pool (to <6 mmol/ kg dry muscle) induced by acetylcarnitine synthesis [39].

As a result, an increase in skeletal muscle total carnitine content may provide more effective buffering of acetyl-CoA production during high-intensity exercise, counteracting the increase in the acetyl-CoA/CoASH ratio and enhancing PDC flow and mitochondrial ATP production [23]. This would diminish the contribution of glycolysis and PCr hydrolysis to ATP synthesis, especially during the rest-to-exercise transition, when inertia in mitochondrial ATP production is known to exist at the level of PDC activation and flux [40]. Furthermore, increasing PDC flow during high-intensity exercise is likely to lower muscle lactate generation, potentially improving exercise performance by reducing muscular acidosis [41]. These metabolic actions could have significant implications for anaerobic ATP during high-intensity exercise. In this way, it has reported improved muscle strength (number of repetitions and third set Lifting Volume) [33] and anaerobic performance by 30s-Wingate test (mean and peak power) [32,33] and decreased perceived exertion effort and an increase in work production [16,23]. Moreover, it has been shown a decreasing post-exercise blood lactate levels after graded exercise test on the treadmill [16]. These data suggest that both chronic and acute L-C supplementation could result in improvements in high intensity performance [16,23,32,33].

In this sense, the main participation of carnitine during the exercise of low intensity, when the PDC activation and the flow are minimum, it is very likely that it is the translocation of mitochondrial fatty acids [23]. Although it has been suggested that free carnitine only limits fat oxidation at exercise intensities greater than 70% of VO2 max, it has been also observed that an acute increase of 15% in the muscle carnitine content reduced glycolytic flow mediated by insulin and PDC activation compared to a control group [22]. In addition, this effect was also followed by an increase in muscle glycogen and long-chain CoA, which indicates an increase in muscle fatty acid oxidation and CHO storage mediated by carnitine [22]. Therefore, free carnitine availability can be a limiting factor in mitochondrial fatty acids translocation both at rest and during low intensity exercise, and an increase in skeletal muscle total carnitine content would increase the fatty acids oxidation while reducing PDC activation and glycogen utilisation during low-moderate intensity exercise [23].

REVIEWER: 2- The authors state that running a Marathon at 4mmol/l is low to moderate intensity (Discussion paragraph 4.2) but this is a clearly heavy intensity not a low or moderate intensity. The current division require a clear justification based on exercise performance (e.g., authors could refer to weight-loss intensities common to be at low or moderate, clinical population training intensities also common to be at low to moderate). However, the studies selected here need to be clearly in the “low to moderate”. Otherwise, authors may consider introducing a 3rd division so they have “low to moderate” “moderate to heavy” and “severe intensity”. I suggest authors to check relevant literature to help authors appreciate such distinction between intensities in a sporting or clinical contexts (keywords: Predictors of exercise performance; Exercise intensity for health and sports performance, oxygen uptake kinetics). Another suggestion is that you consider two divisions (Below HIIT and HIIT), then you can define those divisions based on specific and consistent criteria.

AUTHORS: Thank you for your recommendation. The Colombani et al., study controlled the “Running time” during a marathon. In this case, although the marathon is a high load exercise (time x intensity), it is very difficult to run above 80% VO2 max (high intensity) for 198 minutes (participants’ mean time in the Colombani et al. study).

On the other hand, before starting the study, Colombani et al. performed the test in order to establish Aerobic-Anaerobic thresho ld (4 mmol). Therefore, this test has been eliminated from the tables, since athletes were not supplemented when they performed it.

Regarding the division, we have maintained the two divisions based on Trinity et al., manuscript where they grouped in these 2 levels: “the SV was lower during high-intensity exercise (80, 90 and 100% of VO2max) compared to moderate-intensity exercise (50, 60 and 70% of VO2 max). In this sense, we have added this sentence in 2.1. Literature search strategies section: “The systematic review of the current scientific literature was undertaken for scientific articles that analysed the effects of L-Carnitine supplementation on high exercise performance (≥80% VO2 max) and moderate intensity (50-79% VO2 max) exercise performance [12].”

Trinity, J.D.; Lee, J.F.; Pahnke, M.D.; Beck, K.C.; Coyle, E.F. Attenuated relationship between cardiac output and oxygen uptake during high-intensity exercise. Acta Physiol. (Oxf). 2012, 204, 362–370, doi:10.1111/J.1748-1716.2011.02341.X.

In this sense, the authors have added a paragraph in limitations section assuming the classification of high and low-moderate intensity exercise performance could not be fully adjusted in function of the criteria  used for that partition.

REVIEWER: 3-As mentioned in the previous round, authors need to distinguish between body composition “fat loss” outcomes (e.g. weight or fat loss, changes in body fat percentage, increase in muscle mass. etc.) which are very different and “fat metabolism” that the authors have already discussed throughout. Reference 4 did not measure or tested the carnitine effects on body composition. I suggest removing the statement referring to carnitine effects on “body composition” since no evidence was provided here on such effects, or make relevant amendments.

AUTHORS: Thank you for your recommendation. To avoid misunderstandings about this mentioned topic, authors have reorganized the 1st paragraph and have paraphrased the next text of Karlic and Lohninger: Rumors that L-carnitine supplementation helped the Italian national soccer team to win the World championship in 1982 contributed immensely to its popularity. The most important claim relates to the role of carnitine in fat metabolism. L-carnitine is often advertised to improve fat metabolism, reduce fat mass, and increase muscle mass. In other words, the substance is portrayed as a “fat burner.”

Karlic, H.; Lohninger, A. Supplementation of L-carnitine in athletes: Does it make sense? Nutrition 2004, 20, 709–715, doi:10.1016/j.nut.2004.04.003

REVIEWER: If the authors wish to add this effect (perhaps in Discussion or in Limitation sections), then it needs to be correctly discussed, and there are others who have discussed this in relation to sports performance e.g. https://clinicalnutritionespen.com/article/S2405-4577(20)30053-X/fulltext) or (https://jissn.biomedcentral.com/articles/10.1186/s12970-020-00377-2).

AUTHORS: Thank you for your recommendation. The authors have eliminated that topic as in the previous comment indicated.

Specific

REVIEWER: -Double-check spellings and grammar throughout: e.g. Line 57 “increases” should be “increasing”. Line 56 “and in addition to” should remove “in addition to” or remove “and”. Line 58 “improving body composition, improving insulin resistance and may decrease…” should be “improving body composition and insulin resistance, and decreasing” Line 551 “systematic”.etc.

AUTHORS: Thank you for your recommendation. The authors have reviewed the text.

REVIEWER: Line 60 sentence too long, remove “since” and start new sentence: “Previous evidence has shown that carnitine supplementation…”

AUTHORS: Thank you for your recommendation. The authors have reviewed the sentence.

REVIEWER: Line 75: “potential multiple effects..”

AUTHORS: Thank you for your recommendation. The authors have corrected it.

REVIEWER: Line 77: Needs rephrasing and to be more specific, e.g. “these mechanisms have been suggested/hypothesised to impact sports performance through increasing and maintaining a high muscle carnitine content”

AUTHORS: Thank you for your recommendation. The authors have rephrased that sentence.

REVIEWER: -Reference 15: My previous comment, about adding reference page number or chapter, still needs addressing since you moved what was reference 10 and made it now reference 15.

AUTHORS: Thank you for your recommendation. The authors have changed that reference by Stephens et al.

REVIEWER: -Line 174: Were “double-blind” studies only included?

AUTHORS: Thank you for your question. Yes, this systematic review only has included “double-blind” studies. (See Line 182)

REVIEWER: -Line 436: Specify which type of performance. Please refer to my comment above for accuracy. Reference 17 and 30 are different from 31.

AUTHORS: Thank you for your recommendation. The authors have rephrased that lines: “Wall et al. [23] revealed that a supplementation with 4 g L-CLT (2.72 L-C) with 160 g of CHO distributed in 2 times per day for 24 weeks improved a 35 % more in the work capacity in a “all-out” test. This test was performed after a 30-minutes at 50 % VO2 max followed immediately by 30 minutes at 80% VO2 max [23]. Authors indicated that this advantage in the “all-out” test was motivated because the group supplemented with carnitine had a 71% more muscle glycogen compared to the control motivated by a savings especially in the 30-minutes at 50% VO2 max test [23].”

REVIEWER: -Discussion paragraph 4.2: See comment above about distinctions between low, moderate, heavy, severe and all-out intensities.

AUTHORS: Thank you for your recommendation. The authors hoping to have solved the problem based on the intensities included by Trinity et al.

REVIEWER: -VO2/VO2max: Insert a Dot above the” V”.

AUTHORS: Thank you for your recommendation. The authors have made this change.

REVIEWER: -Line 644: Remove “Patents”, just keep it as “Author contribution”

AUTHORS: Thank you for your recommendation. The authors have deleted “Patents”.
